

# HEPPA-II model-measurement intercomparison project: EPP indirect effects during the dynamically perturbed NH winter 2008–2009

Bernd Funke[1], William Ball[2], Stefan Bender[4], Angela Gardini[1], V. Lynn Harvey[5], Alyn Lambert[6], Manuel López-Puertas[1], Daniel R. Marsh[7], Katharina Meraner[8], Holger Nieder[4], Sanna-Mari Päivärinta[3,9], Kristell Pérot[10], Cora E. Randall[5], Thomas Reddmann[4], Eugene Rozanov[2,11], Hauke Schmidt[8], Annika Seppälä[3], Miriam Sinnhuber[4], Timofei Sukhodolov[2], Gabriele P. Stiller[4], Natalia D. Tsvetkova[12], Pekka T. Verronen[3], Stefan Versick[4,14], Thomas von Clarmann[4], Kaley A. Walker[13], and Vladimir Yushkov[12]

[1]Instituto de Astrofísica de Andalucía, CSIC, Apdo. 3004, 18008 Granada, Spain.
[2]Physikalisch-Meteorologisches Observatorium, World Radiation Center, Davos, Switzerland
[3]Earth Observation Unit, Finnish Meteorological Institute, Helsinki, Finland
[4]Karlsruhe Institute of Technology (KIT), Institute of Meteorology and Climate Research (IMK-ASF), P.O. Box 3640, 76021 Karlsruhe, Germany.
[5]Laboratory for Atmospheric and Space Physics, University of Colorado, Boulder, USA
[6]Jet Propulsion Laboratory, California Institute of Technology, Pasadenia, California, USA
[7]National Center for Atmospheric Research, Boulder, Colorado, USA
[8]Max Planck Institute for Meteorology, Hamburg, Germany
[9]Department of Physics, University of Helsinki, Helsinki, Finland
[10]Chalmers University of Technology, Göteborg, Sweden
[11]Institute for Atmospheric and Climate Science ETH, Zurich, Switzerland
[12]Central Aerological Observatory, Moscow region, Russia
[13]Department of Physics, University of Toronto, Toronto, Ontario, Canada
[14]Karlsruhe Institute of Technology (KIT), Steinbuch Centre for Computing (SCC), Karlsruhe, Germany.

*Correspondence to:* B. Funke (bernd@iaa.es)

**Abstract.**

We compare simulations from three high-top (with upper lid above 120 km) and five medium-top (with upper lid around 80 km) atmospheric models with observations of odd nitrogen ($NO_x = NO + NO_2$), temperature, and carbon monoxide from seven satellite instruments (ACE-FTS on SciSat, GOMOS, MIPAS, and SCIAMACHY on Envisat, MLS on Aura, SABER on TIMED, and SMR on Odin) during the Northern Hemisphere (NH) polar winter 2008/2009. The models included in the comparison are the 3d Chemistry Transport model (3dCTM), the ECHAM5/MESSy Atmospheric Chemistry (EMAC) model, FinROSE, the Hamburg Model of the Neutral and Ionized Atmosphere (HAMMONIA), the Karlsruhe Simulation Model of the Middle Atmosphere (KASIMA), the modeling tools for SOlar Climate Ozone Links studies (SOCOL and CAO-SOCOL), and the Whole Atmosphere Community Climate Model (WACCM4). The comparison focuses on the energetic particle precipitation (EPP) indirect effect, that is, the polar winter descent of $NO_x$ largely produced by EPP in the mesosphere and lower thermosphere. A particular emphasis is given to the impact of the sudden stratospheric warming (SSW) in January 2009 and





the subsequent elevated stratopause (ES) event associated with enhanced descent of mesospheric air. The chemistry climate model simulations have been nudged toward reanalysis data in the troposphere and stratosphere while being unconstrained above. An odd nitrogen upper boundary condition obtained from MIPAS observations has further been applied to medium-top models. Most models provide a good representation of the mesospheric tracer descent in general, and the EPP indirect effect

in particular, during the unperturbed (pre-SSW) period of the NH winter 2008/2009. The observed $NO_x$ descent into the lower mesosphere and stratosphere is generally reproduced within 20%. Larger discrepancies of a few model simulations could be traced back either to the impact of the models' gravity wave drag scheme on the polar wintertime meridional circulation or to a combination of prescribed $NO_x$ mixing ratio at the uppermost model layer and low vertical resolution. In March–April, after the ES event, however, modelled mesospheric and stratospheric $NO_x$ distributions deviate significantly from the obser-

vations. The too fast and early downward propagation of the $NO_x$ tongue, encountered in most simulations, coincides with a temperature high bias in the lower mesosphere (0.2–0.05 hPa) being likely caused by an overestimation of descent velocities. On the other hand, upper mesospheric temperatures (at 0.05–0.001 hPa) are generally underestimated by the high-top models after the onset of the ES event, being indicative for too slow descent and hence too low $NO_x$ fluxes. As a consequence, the magnitude of the simulated $NO_x$ tongue is generally underestimated by these models. Descending $NO_x$ amounts simulated

with medium-top models are on average closer to the observations but show a large spread of up to several hundred percent. This is primarily attributed to the different vertical model domains in which the $NO_x$ upper boundary condition is applied. In general, the intercomparison demonstrates the ability of state-of-the-art atmospheric models to reproduce the EPP indirect effect in dynamically and geomagnetically quiescent NH winter conditions. The encountered differences between observed and simulated $NO_x$, CO, and temperature distributions during the perturbed phase of the 2009 NH winter, however, emphasize

the need for model improvements in the dynamical representation of elevated stratopause events in order to allow for a better description of the EPP indirect effect under these particular conditions.

## 1 Introduction

The potential impact of energetic particle precipitation (EPP) on regional climate is nowadays becoming recognized. Solar forcing recommendations for the recently launched Climate Model Intercomparison Project Phase 6 (Eyring et al., 2016)

include, for the first time, the consideration of energetic particle effects (Matthes et al., 2016). EPP is strongly linked to solar activity and hence to the solar cycle, either directly by coronal mass ejections producing sporadically large fluxes of solar energetic particles or indirectly by the quasi-continuous impact of the solar wind on the Earth's magnetosphere. In the mesosphere and lower thermosphere (MLT), EPP-induced ionisation initiates the production of odd nitrogen and odd hydrogen (the latter below ∼85 km), both of them destroying ozone via catalytic cycles. Odd nitrogen ($NO_x = NO + NO_2$) is long-lived

during polar winter and is then regularly transported down from its source region into the stratosphere, to altitudes well below 30 km (e.g., Randall et al., 2007; Funke et al., 2014a). This so-called EPP indirect effect contributes significant amounts of $NO_x$ to the polar middle atmosphere during each winter. EPP-induced ozone changes are thought to modify the thermal structure and winds in the stratosphere which, in turn, modulate the strength of the Arctic polar vortex. The introduced signal could then



propagate down to the surface, introducing significant variations of regional climate, particularly in the Northern Hemisphere (NH) (Seppälä et al., 2009; Baumgaertner et al., 2011; Rozanov et al., 2012; Seppälä and Clilverd, 2014; Maliniemi et al., 2014).

At present, many chemistry climate models account for EPP-induced ionization and its chemical impact on the neutral at-
mosphere, which is required for the simulation of atmospheric EPP effects and ultimately for the investigation of potential EPP-climate links. A comprehensive evaluation of these models' capacity to reproduce observed EPP effects by means of coordinated intercomparison studies is a necessary step towards this goal. The High Energy Particle Precipitation in the Atmosphere (HEPPA) model vs. data intercomparison initiative (Funke et al., 2011) evaluated the chemical response of nitrogen and chlorine species in nine atmospheric models to the "Halloween" solar proton event in late October 2003 with observations taken
by the Michelson Interferometer for Passive Atmospheric Sounding (MIPAS) on Envisat. Reasonable agreement of observed and modelled reactive nitrogen and ozone changes was found, demonstrating the models' overall ability to reproduce the direct EPP effect by solar protons. However, most models failed to adequately describe the repartitioning of nitrogen compounds in the aftermath of the event which could be attributed to deficiencies in the representation of the D-region ion chemistry and motivated recent model developments (Egorova et al., 2011; Verronen et al., 2016; Andersson et al., 2016).

The observation-based evaluation of the simulated atmospheric effects of magnetospheric particles, which are thought to be of higher relevance for climate, is more challenging because of the quasi-continuous flux of electrons compared to protons, and the difficulty in separating between local production and downward transport of $NO_x$ during polar winter. Although a pronounced dependence of reactive nitrogen enhancements in the polar winter stratosphere and mesosphere on the geomagnetic activity levels has been demonstrated (Funke et al., 2014b), dynamical variability, particularly in the NH, can mask out this
effect. In particular, the occurrence of elevated stratopause (ES) events following sudden stratospheric warmings (SSWs) during Arctic winters often causes much larger mesospheric $NO_x$ enhancements than expected from the actual geomagnetic activity level, after a brief $NO_x$ depletion related to the weakened vortex during the SSW. The ability of climate models to adequately simulate tracer transport in Arctic winters, including perturbed winters characterized by SSW/ES events, is therefore crucial to accurately model EPP effects and their possible NH regional climate impacts.

Simulations of mesospheric tracer descent during dynamically perturbed NH winters have been compared with observations in several studies. Using the KArlsruhe SImulation Model of the middle Atmosphere (KASIMA) with specified dynamics below 48 km and prescribed $NO_x$ concentrations from MIPAS night time $NO_2$ observations above 55 km, Reddmann et al. (2010) calculated the amount of EPP-$NO_x$ entering the stratosphere from July 2002 to March 2004. KASIMA reproduced the MIPAS observations of $NO_x$ entering the stratosphere reasonably well, even during the SSW winter 2003/2004. However,
the ability of the model to adequately simulate mesospheric tracer transport could not be tested because of the constrained $NO_x$ in the mesosphere. Salmi et al. (2011) and Päivärinta et al. (2016), in turn, used FinROSE with constrained $NO_x$ at the upper boundary (∼80 km) for both early 2009 and 2012. Their results show that FinROSE is able to qualitatively reproduce the downward descent of $NO_x$ from the MLT region into the stratosphere, but the actual $NO_x$ amounts can differ significantly from the observations. In the case of CTMs, the results are strongly affected by the meteorological data, i.e., a source of uncer-
tainty, used to drive the model. McLandress et al. (2013) used a version of the Canadian Middle Atmosphere Model (CMAM)



that was nudged toward reanalysis data up to 1 hPa to examine the impacts of parametrised orographic and non-orographic gravity wave drag (GWD) on the zonal mean circulation of the mesosphere during the perturbed NH winters 2006 and 2009 in comparison with temperature and carbon monoxide (CO) observations from the Microwave Limb Sounder (MLS) on Aura. They found that non-orographic GWD is primarily responsible for driving the circulation that results in the descent of CO from the thermosphere following the warmings. Randall et al. (2015) investigated the $NO_x$ descent during the Arctic winter/spring of 2004 with Whole Atmosphere Community Climate Model (WACCM) simulations that were nudged to Modern-Era Retrospective Analysis for Research and Applications (MERRA) data. They found that their simulated $NO_x$, although qualitatively reproducing the enhanced descent after the ES event, was up to a factor of 5 too low compared with satellite observations. This underestimation was attributed to missing NO production by high-energy electrons in the mesosphere in combination with an underestimation of mesospheric descent during the recovery phase after the SSW. Siskind et al. (2015) compared simulations of mesospheric tracer descent in the winter and spring of 2009 with two versions of WACCM, one constrained with data from MERRA which extends up to 50 km and the other constrained to the Navy Operational Global Atmospheric Prediction System-Advanced Level Physics High Altitude (NOGAPS-ALPHA) which extends up to 92 km. By comparison with Solar Occultation for Ice Experiment (SOFIE) data they showed that constraining WACCM to NOGAPS-ALPHA yields a dramatic improvement in the simulated descent of enhanced $NO_x$ and very low methane.

Most of these studies suggest that the model representation of the perturbed dynamics during NH winters with SSWs and ES events has a crucial impact on the simulated amount of $NO_x$ transported into the stratosphere and that a proper parametrisation of unresolved GWD is key to achieving agreement with observations. However, previous studies focused on individual models, making it difficult to assess the overall ability of state-of-the-art atmospheric models to reproduce the EPP indirect effect in NH winters. Comprehensive multi-model intercomparisons addressing dynamically perturbed NH winters, however, have so far been restricted to the assessment of the temperature zonal mean, planetary wave, and tidal variability during the 2009 SSW event in the middle and upper atmosphere (Pedatella et al., 2014), as well as on the impacts on the ionosphere variability (Pedatella et al., 2016). Further, although our knowledge of temperature and tracer distributions in polar winters has dramatically increased with the advent of atmospheric satellite observations, specific intercomparisons and validation efforts focussing on such conditions are sparse. A systematic assessment of this knowledge is therefore essential to quantitatively diagnose the model performance with respect to mesospheric tracer transport under perturbed (and unperturbed) polar winter conditions.

A coordinated intercomparison project focussing on tracer descent and the EPP indirect effect during such a winter was therefore initiated in the frame of the SPARC/WCRP's SOLARIS-HEPPA activity. In this so-called HEPPA-II project, simulations of the NH polar winter 2008/2009 from eight atmospheric models have been compared with observations of temperature and concentrations of $NO_x$ and CO from seven satellite instruments including the Atmospheric Chemistry Experiment Fourier Transform Spectrometer (ACE-FTS) on SciSat, the Envisat instruments Global Ozone Monitoring by Occultation of Stars (GOMOS), MIPAS, and the SCanning Imaging Absorption spectroMeter for Atmospheric CHartographY (SCIAMACHY), as well as MLS on Aura, the Sounding of the Atmosphere using Broadband Emission Radiometry (SABER) instrument on the Thermosphere, Ionosphere, Mesosphere, Energetics and Dynamics (TIMED) satellite, and the Sub-Millimetre Radiometer (SMR) on Odin. The 2008/2009 winter was chosen for this intercomparison exercise not only because of its peculiar dynami-





cal conditions, characterized by the pronounced SSW in January and the unusually strong descent of odd nitrogen despite the low geomagnetic activity level around solar minimum, but also because of the availability of a large number of observations from different satellite instruments that allowed for a detailed evaluation of the model simulations. The models included in the comparison are the 3d Chemistry Transport model (3dCTM), the ECHAM5/MESSy Atmospheric Chemistry (EMAC) model,

FinROSE, the Hamburg Model of the Neutral and Ionized Atmosphere (HAMMONIA), KASIMA, the modeling tools for SOlar Climate Ozone Links studies (SOCOL and CAO-SOCOL), and WACCM (Version 4). Only three of these models (3dCTM, HAMMONIA, and WACCM) extend up into the lower thermosphere where a large fraction of EPP-induced odd nitrogen production occurs. All other models have their upper lid in the mesosphere and require an odd nitrogen upper boundary condition, accounting for EPP production higher up, for simulating the introduced EPP indirect effect in the model domain. This upper

boundary condition (UBC) has been constructed from $NO_x$ observations of the MIPAS instrument taken during the Arctic winter 2008-2009.

In this study we report results from the HEPPA-II intercomparison project. A major aim is the identification and characterisation of model biases and their uncertainties in the simulations of the perturbed 2008/2009 NH winter by systematically comparing to the suite of satellite observations. For this purpose, common diagnostics are applied in all comparisons, and the

sampling characteristics of the instruments are taken into account. Since the study focusses on the evaluation of the ability of the models to simulate the source and transport of MLT tracers by means of observed quantities (i.e., temperature and trace gas abundances), any more sophisticated analysis, e.g., qualifying the different GW drag parametrisations etc., is outside the scope of this comparison. On the other hand, our analysis should motivate such studies to identify the deficits in key processes of this vertical coupling.

The paper is organised as follows: Section 2 gives an overview on the satellite observations and data products used in this study. Section 3 describes the participating chemistry climate and transport models. The $NO_x$ UBC employed in the medium-top models is described in Sec. 4, and Sec. 5 introduces the intercomparison method. Results of the intercomparisons are discussed in Sec. 6 with focus on the representation of the EPP indirect effect by the high-top models in the upper mesosphere and lower thermosphere and, in Sec. 7, with focus on the upper stratospheric and mesospheric representation in all models.

## 2  Satellite observations

### 2.1  ACE-FTS/SciSat

The ACE-FTS has performed infrared solar occultation measurements from the SciSat satellite since 21 February 2004 following its launch on 12 August 2003 (Bernath et al., 2005). The instrument is a Fourier transform spectrometer operating between 750 and 4400 cm$^{-1}$ with a spectral resolution of 0.02 cm$^{-1}$. The SciSat satellite is in a highly-inclined circular orbit (74°)

and thus provides measurements from 85 °N to 85 °S over each year with a significant focus on polar measurements. Up to 30 measurements are made each day by ACE-FTS extending from the cloud tops to ∼150 km and, from these sets of spectra, profiles of over 30 trace gases and isotopologues, temperature and pressure are retrieved. For this study, version 3.0 of the ACE-FTS data set was used which covers 21 February 2004 to 30 September 2010.





The ACE-FTS retrieval algorithm is described in Boone et al. (2005) and the specific details of version 3.0/3.5 are provided in Boone et al. (2013). Briefly, an unconstrained non-linear least squares global fitting approach is used to fit the measured and forward modelled spectra. Selected $CO_2$ lines in the spectra are used to retrieve pressure and temperature as a function of altitude and then these results are used to retrieve volume mixing ratio (VMR) profiles of the various trace gases from

microwindows selected for each of the target molecules. The vertical resolution of the ACE-FTS measurements is ∼3 km, based on the instrument field-of-view (Boone et al., 2005). $NO_x$ is provided from ACE-FTS using the retrieved NO (6–107 km) and $NO_2$ (7–52 km) profiles. Above 52 km, where both sunset and sunrise $NO_2$ concentrations are very small and hence not detectable, the scaled a priori $NO_2$ profile has been used to extend the $NO_x$ profiles to the higher altitudes. The CO profiles extend from 5–110 km and temperature is retrieved from 15–125 km.

The version 3.5 NO profiles differ from HALOE by -15 to 6% between 27 and 53 km and from summertime MIPAS measurements by -9 to 2% between 36 and 52 km (Sheese et al., 2016a). For $NO_2$, the bias found between ACE-FTS and a suite of other limb and occultation sounders is better than 18% from 17-27 km and -15% from 28-41 km (Sheese et al., 2016a). For both of these species, a box model was used to apply a diurnal scaling to the ACE-FTS profiles before the comparisons. ACE-FTS CO has been compared with MIPAS and MLS by Sheese et al. (2016b). On average, there is a -11% bias between

28 and 50 km with respect to MIPAS and a bias of ±10%. Based on comparisons with coincident satellite observations (within 350 km and 3 hours), it has been found that ACE-FTS v3.5 temperatures agree to within ±2 K between 15 and 40 km, within ±7 K between 40 and 80 km and within ±12 K between 80 and 100 km (P. Sheese, personal communication).

## 2.2 GOMOS/Envisat

GOMOS was a stellar occultation instrument on the polar orbiting Envisat satellite, operating between 2002–2012 (Bertaux

et al., 2010). This satellite has been flying in a sun-synchronous orbit at approximately 800 km altitude and crossing the equator at 10:00/22:00 local time. Unfortunately, the communication to the satellite was lost in April 2012. GOMOS consisted of a UV-visible spectrometer for wavelengths 250–675 nm, two IR channels, and two photometers, measuring the stellar flux through the atmosphere at high sampling frequency. GOMOS measured vertical profiles of $O_3$, $NO_2$, $NO_3$, $H_2$, O, $O_2$, and aerosols in the middle atmosphere. The altitude range varies for each constituent. For example, for ozone the altitude range is about

16–100 km, whereas for $NO_2$ the altitude range in non-polar conditions is 20–50 km and extends up to 70 km in polar winter when enhanced amounts of $NO_2$ are present in the atmosphere (Seppälä et al., 2007; Hauchecorne et al., 2007). The altitude sampling resolution of GOMOS varies between 0.4 km and 1.7 km depending on measurement geometry (Tamminen et al., 2010). After application of Tikhonov type smoothing, the target resolution of the ozone observations becomes 2 km below 30 km and 3 km above 40 km. For all other gases the target resolution is 4 km (Kyrölä et al., 2010).

Here, we have used GOMOS $NO_2$ profiles (version GOPR_6.0c_6.0f) measured in night time conditions (solar zenith angle at tangent point location > 107°; solar zenith angle at spacecraft location > 90° to avoid stray light). In addition to the night time condition, only occultations where the temperature of the star was > 6000 K were selected. This mainly influences the precision of the ozone observations, while $NO_2$ is less affected. The typical precision of the $NO_2$ measurements is 5-20% while the systematic error of the $NO_2$ observations is estimated to be of the order of few percent (1-5%) (Tamminen et al.,



2010; Verronen et al., 2009). As NO is quickly converted into $NO_2$ by reaction with $O_3$ after sunset, the night time GOMOS $NO_2$ measurements used here are a reasonable representation of stratospheric and lower mesospheric $NO_x$.

Because stars are used as the light source, the locations of the observations change with time. A representative distribution of the latitudes sampled during the course of a year can be seen in Figs. 7–9 of Bertaux et al. (2010). Due to this sampling, for

the NH polar region in winter 2008–2009, GOMOS night time $NO_2$ observations were available for the period of December 2008–January 2009. GOMOS measurements provide the constituent profiles as number densities. For the purpose of this study these were converted to volume mixing ratios using temperature and pressure profiles from the WACCM model (see below).

### 2.3 MIPAS/Envisat

The MIPAS instrument (Fischer et al., 2008) on Envisat provided global stratospheric and mesospheric measurements of tem-

perature (García-Comas et al., 2014), NO and $NO_2$ (Funke et al., 2014a), CO (Funke et al., 2009), as well as numerous other trace species (e.g., von Clarmann et al., 2009, 2013) during 2002–2012. Here, we use observations taken in the nearly continuous nominal observation mode (scanning range 6–70 km, hereinafter referred to as MIPAS-NOM), as well as occasional special mode observations (middle and upper atmospheric observation modes covering 20–100 km and 40–170 km, respectively, hereinafter referred to as MIPAS-UA), the latter taken with a frequency of about 1 out of 5 days. We also use special

mode UA observations which include three orbits per day passing the 20°W–70°E and 160°E–110°W sectors during 14-–18 January and 21—27 January 2009 and which were taken as support for the Dynamics and Energetics of the Lower Thermosphere in Aurora 2 (DELTA-2) campaign (Abe et al., 2006). MIPAS passes the equator in a southerly direction at 10:00 a.m. local time 14 to 15 times a day, observing the atmosphere during day and night with global coverage from pole to pole and a horizontal along-track sampling between 275 and 410 km depending on the observation mode.

Temperature and trace gas profiles have been retrieved from calibrated geolocated limb emission spectra with the scientific MIPAS level 2 processor developed and operated by the Institute of Meteorology and Climate Research (IMK) in Karlsruhe together with the Instituto de Astrofísica de Andalucía (IAA) in Granada. The general retrieval strategy, which is a constrained multi-parameter non-linear least squares fitting of measured and modelled spectra, is described in detail in von Clarmann et al. (2003). Its extension to retrievals under consideration of non-LTE (i.e. CO, NO, and $NO_2$) is described in Funke et al. (2001).

Non-LTE vibrational populations of these species are modeled with the Generic RAdiative traNsfer AnD non-LTE population Algorithm (GRANADA) (Funke et al., 2012) within each iteration of the retrieval.

MIPAS-NOM $NO_x$ data have been built from NO and $NO_2$ data versions V5r_NO_220 and V5r_NO2_220, respectively. MIPAS-UA $NO_x$ data is based on data versions V4o_NO_501/611 and V4o_NO2_501/600. In the middle- to high-latitude polar winters, typical vertical resolutions are 4—6 km below 50 km and 6—9 km above, while the single measurement precision

is on the order of 5–15%. Systematic errors, dominated by non-LTE-related uncertainties of NO and $NO_2$, have been estimated to be less than 10%. CO data (version V5r_CO_220) used here have a single measurement precision ranging from 20–30% above 45 km to 70–80% in the lower stratosphere. The vertical resolution is 6—12 km. The single measurement precision of temperature data (versions v5r_T_220 and v5r_T_521/621 for MIPAS-NOM and MIPAS-UA, respectively) is 0.5—2 K





below 70 km and 2—7 K above. The systematic error is typically 1—3 K below 85 km and 3–11 K above. The average vertical resolution is 3—6 km below 90 km, and 6—10 km above.

## 2.4 MLS/Aura

The Microwave Limb Sounder (MLS) instrument (Waters et al., 2006) was launched on 15 July 2004 and measures thermal
microwave emission from Earth's limb. On each day MLS provides ∼3500 vertical profiles of temperature and trace gases between 82°S and 82°N spaced ∼1.5° apart along great circles following the orbit track. Version 4.2 temperature and CO are used here. Temperature is deemed useful for scientific studies between 316 hPa and 0.001 hPa. The vertical resolution is 5 km near 40 km and increases to ∼10 km near 90 km (Livesey, 2016). In the mesosphere, systematic and random errors are 2.5 K and comparisons with correlative measurements show a 0–7 K cold bias (Schwartz et al., 2008). CO is recommended for
scientific use from 215 hPa to 0.0046 hPa (Pumphrey et al., 2007). The vertical resolution is 4–5 km in the stratosphere and 6-7 km in the mesosphere. Froidevaux et al. (2006) indicate that the CO data have a 25–50% positive bias in the mesosphere. Estimates of absolute accuracy are 10% (Filipiak et al., 2005). For this work, version 4.2 temperature and CO data have been filtered using the precision, status, quality, and convergence values provided by the MLS science team (Livesey, 2016).

## 2.5 SABER/TIMED

The SABER (Sounding of the Atmosphere using Broadband Emission Radiometry) instrument is a 10 channel limb scanning
radiometer. It scans the Earth's limb continuously recording profiles of infrared radiance from the atmosphere in discrete spectral intervals (Russell et al., 1999). The specific wavelength bands observed by SABER were chosen so that a variety of data products could be retrieved or derived, including kinetic temperature, ozone, water vapor, carbon dioxide, atomic oxygen, atomic hydrogen, rates of energy deposition, rates of energy loss, and rates of radiative heating and cooling. The instrument was
launched onboard the NASA Thermosphere-Ionosphere-Mesosphere Energetics and Dynamics (TIMED) mission in December 2001. The TIMED satellite is in an orbit inclined 74° with respect to the equator. In order to keep SABER within its allowable range of operating temperatures, the spacecraft executes a rotation about its yaw axis every 60 days. While in the "northward" viewing mode the instrument views from approximately 83°N to 52°S, after the yaw maneuver the instrument views from approximately 52°N to 83°S. SABER began acquiring scientific data in January 2002 and is still operating with nearly 100%
duty cycle. Here we use the temperature measured from 1 October 2008 until 31 May 2009. In that period SABER was measuring in the "northward" viewing mode during the sub-periods: 1 October–17 November in 2008; 11 January–15 March and 18 May–31 May in 2009. The rest of the days, i.e., 17 November 2008–15 January 2009 and 15 March–19 May in 2009, it was observing in the "southward" viewing mode. During those 60-day yaw periods SABER data are available during day and nighttime at almost all local times.

SABER's temperatures are derived from measurements of the $CO_2$ limb radiance at 15 $\mu$m, in the altitude range from 20 to 100 km, and thus require non-LTE calculations in the retrievals. The inversion of temperature and pressure (version 1.07) are described in detail by Remsberg et al. (2008). SABER vertical sampling is 400 m and its vertical resolution is about 2 km. Typical SABER single measurement estimates of v1.07 temperature random errors are <0.5 K below 55 km, 1 K at 70 km, 2 K



at 85 km and 7 K at 100 km. The systematic errors are $<1.5$ K below 55 km, 0.5 K at 70 km, 4 K at 85 km and 5 K at 100 km (Remsberg et al., 2008; García-Comas et al., 2008). This study uses data from the Level 2A files of version 2.0. Nevertheless, the v2.0 estimated systematic and random errors are not expected to change much from v1.07 since the uncertainties of their sources are the same. A thorough comparison of these temperatures with those measured by other satellites, MIPAS, ACE-FTS,

MLS, OSIRIS, SOFIE and by lidar measurements, has been recently carried out by García-Comas et al. (2014) in the study about the validation of MIPAS vM21 temperatures. The comparison of SABER v2.0 with MIPAS vM21 is remarkably good, with differences smaller than 2 K at all altitudes and seasons, except for high-latitude summers above 65 km where they are 3–4 K at 65-80 km (MIPAS colder) and 5-7 K around the mesopause (MIPAS warmer).

## 2.6  SMR/Odin

The SMR (Sub-Millimetre Radiometer) instrument is a limb emission sounder aboard Odin, a Swedish-led satellite launched in 2001 in cooperation with the Canadian, French and Finnish space agencies (Murtagh et al., 2002). It was initially a joint astronomy and aeronomy mission and, until 2007, the observation time was equally divided between the two disciplines. The satellite became a European Space Agency (ESA) third-party mission in 2007, and is entirely dedicated to atmospheric measurements since the same date. Odin is orbiting the Earth in a sunsynchronous orbit at an initial altitude of 580 km and at

Equator-crossing times varying between 06:00 and 07:00 am/pm local time. These parameters are slightly changing with time due to the drifting orbit.

SMR is measuring globally a variety of trace gases and the temperature from the upper troposphere to the lower thermosphere. For this purpose, it uses four sub-millimetre channels (486.1–503.9, 541.0–558.0, 547.0–564.0, 563.0–581.4 GHz) and one millimetre-wave channel (118.25–119.25 GHz) (Merino et al., 2002). The observation of different species requires

switching from one channel to another.

Nitric oxide is retrieved from the observation of thermal emission lines in a band centred around 551.7 GHz. The version 2.1 of NO data is used in this study. The overall vertical coverage is from 7 to 115 km, and in the altitude range considered here the vertical resolution is about 7 km (Pérot et al., 2014). NO data is available approximately four days per month after 2007, on an irregular basis of two observation days in a 14-day cycle. Systematic errors amount to 3% from spectroscopic parameters, 2%

from calibration, and 3–6% from sideband suppression (Sheese et al., 2013). The single measurement retrieval error amounts to 44-48%, in the case of Antarctic night time mesosphere-lower thermosphere, as studied by Sheese et al. (2013). A comparison study performed by Bender et al. (2015) showed that SMR NO measurements were consistent with NO measurements by SCIAMACHY, MIPAS and ACE-FTS, despite the different measurement methods and retrieval strategies used for these four instruments.

## 30   2.7  SCIAMACHY/Envisat

The SCanning Imaging Absorption spectroMeter for Atmospheric CHartographY (SCIAMACHY, see Burrows et al. (1995); Bovensmann et al. (1999)) is a limb-sounding UV–vis–NIR spectrometer on Envisat. SCIAMACHY comprises eight spectral channels from 214 nm to 2380 nm with spectral resolutions ranging from 0.22 nm to 1.48 nm. Among the main measurement





modes, the nominal limb mode carried out limb measurements from ground to 105 km until mid-October 2003. After 15 October 2003, the nominal mode was restricted to 91 km top altitude. From July 2008 until the end of Envisat in April 2012, SCIAMACHY carried out a special mesosphere–lower thermosphere mode (MLT), scanning from 50 km to 150 km one day every two weeks. The average horizontal distance between the individual limb scans was about seven degrees in both cases.

Nitric oxide is retrieved from the NO gamma bands observed with SCIAMACHY's UV channel 1 (230–314 nm) (Bender et al., 2013, 2016). The tomographic orbit retrieval was carried out from 60 km to 160 km and from 90°S to 90°N on a fixed 2 km×2.5° altitude–latitude grid. The retrieval from the MLT mode delivered the NO number densities with a vertical resolution of 5–10 km at altitudes from 70 km to 150 km. With the nominal mode, this resolution is achieved between 65 km and 80 km. The average single orbit measurement error amounts to about 30%. Systematic errors amount to 7% from uncertain

spectroscopic data, 3% from uncertainties in the solar spectrum (Chance and Kurucz, 2010), and about 10% from temperature uncertainties. As the NO gamma bands are excited by absorption of solar light, the retrieval of NO is restricted to daylight observations. Polar winter data are therefore restricted to latitudes equatorward of the polar night terminator (around 70° in the mesosphere/lower thermosphere at winter solstice).

The retrieved NO number densities from the MLT mode have been compared to ACE-FTS, MIPAS, and SMR (Bender

et al., 2015). The measurements were found to be consistent among all instruments with SCIAMACHY retrieving slightly lower densities compared to the other instruments during polar winter but higher values in mesospheric polar summer and mid-to-low latitudes.

## 3    Chemistry climate models

In the following, the participating atmospheric models are described and details on the setup of the simulations are provided.

Since the dynamical evolution in the mesosphere is strongly constrained by the behaviour of the lower atmosphere, particularly during a perturbed NH winter, model simulations have been either nudged to or rely entirely on meteorological reanalysis data in order to allow for comparisons to observations. High-top models, having their upper lid above 120 km and including explicit schemes for consideration of $NO_x$ production by particle induced ionization, are described in Sec. 3.1. Medium-top models, having their upper lid around 80 km, are described in Sec. 3.2. These models applied a common odd nitrogen UBC

in order to account for EPP production above the model domain (see Sec. 4). A summary of the different model settings and characteristics is given in Table 1.

### 3.1    High-top models

### 3dCTM

3dCTM is a global 3-dimensional chemistry-transport model developed based on the chemistry scheme of the SLIMCAT model

(Chipperfield, 1999) and the transport scheme of the CTM-B Sinnhuber et al. (2003) for use in the middle atmosphere up to the lower thermosphere. It runs on fixed pressure surfaces from 300 hPa (about 10 km) to $4.96 \times 10^{-6}$ hPa (about 150 km), with





**Table 1.** Summarized description of the atmospheric models involved in this study.

| High-top model | vertical domain (km) | horizontal resolution | vert. res. (km) | meteorological data nudging[a] | family approach[a] | kinetic data[b] | EPP-NO$_x$ production |
|---|---|---|---|---|---|---|---|
| 3dCTM | $\sim$10–150 | 2.5°×3.75° | $\sim$1–3 | LIMA (ECMWF < 1 hPa) | no | S06 | AIMOS 1.2 |
| HAMMONIA | $\sim$0–250 | 1.9°×1.9° | $\sim$3 | ERA-I (<1 hPa) | no | S06 | AIMOS 1.6 |
| WACCM | $\sim$0–140 | 1.9°×2.5° | $\sim$1.5 | MERRA (<50 km) | no | S11 | auroral prod. |
| Medium-top model | | | | | | | NO$_x$ UBC range (hPa) |
| CAO-SOCOL | $\sim$0–80 | 3.75°×3.75° | $\sim$2 | ERA-I (<1 hPa) | no | S06 | 0.01 |
| FinROSE | $\sim$0–80 | 6°×3° | $\sim$2–7 | ECMWF (whole model domain) | no | S06 | 0.03–0.01 |
| KASIMA | $\sim$7–120 | 2.8°× 2.8° | 0.75–3.8 | ERA-I (<1 hPa) | no | S03 | 0.03 |
| EMAC | $\sim$0–80 | 2.8°×2.8° | $\sim$1–4 | ERA-I (<0.2 hPa) | reduced | S11 | 0.09–0.01 |
| SOCOL | $\sim$0–80 | 2.8°×2.8° | $\sim$2 | ERA-I (<1 hPa) | no | S11 | 0.01 |

[a] see model descriptions in Sec. 3 for details.

[b] S11: Sander et al. (2011b), S03: Sander et al. (2003), S06: Sander et al. (2006),

a vertical spacing ranging from 1 km in the lower stratosphere and at the mesopause, to about 3 km in the upper stratosphere, lower mesosphere, and lower thermosphere. The horizontal resolution is about 3.75° in longitudinal, about 2.5° in latitudinal direction. Temperature as well as horizontal and vertical wind fields are prescribed by data from the LIMA general circulation model (Berger, 2008), and the model upper boundary is defined by the availability of these data. For the version used here, LIMA is nudged to (1°×1°) ECMWF operational data with a constant nudging of temperature, zonal and meridional winds between the surface and 35 km, and a linear decrease in nudging strength to 45 km, the upper limit of the nudging area. No parametrisation of the gravity wave drag is implemented either in LIMA or in 3dCTM. Only waves with horizontal scales of $\geq$ 500 km and a temporal resolution of 2–12 hours are represented Berger (2008). A comparison of momentum flux climatologies provided in Figure 7 of Berger (2008) with common gravity wave drag schemes as shown, e.g., in Figure 5 of Holton and Zhu (1984), shows that the gravity wave momentum flux in the mesosphere is underestimated by LIMA by about a factor of 2–3 in both the summer and winter hemisphere. In the winter hemisphere, also the vertical structure of the GW momentum flux is somehow different; while Holton and Zhu (1984) essentially show one broad peak at $\sim$65–95 km altitude, varying in strength from -80–120 ms$^{-1}$d$^{-1}$, the LIMA profile shows a double peak structure with a broad peak of -40–60 ms$^{-1}$d$^{-1}$ at $\sim$70–90 km altitude, a minimum in 90–100 km, and a secondary peak above 100 km. This means that the vertical downward motion throughout the mesosphere will be underestimated during winter.

The model chemistry scheme is based on the JPL-2006 recommendation (Sander et al., 2006) but has been adapted from the original SLIMCAT code for use in the mesosphere and lower thermosphere as described in Sinnhuber et al. (2012): the





model considers the photolysis of $O_2$, $CO_2$, $CH_4$, and $H_2O$ in the far-UV wavelength range down to the Lyman $\alpha$ line. Also, in the mesosphere and lower thermosphere, chemical families are not considered for $NO_x$ and $O_x$ species, and $H_2O$, $O_2$, and $H_2$ are now integrated as active chemical species in the model. Additionally, parametrisations for the impact of atmospheric ionization from particle impact and photo-ionisation are considered based on ion-chemistry model studies (Nieder et al., 2014).

The photo-ionisation rate is based on the parametrisation of Solomon and Qian (2005); particle impact ionization rates are prescribed using the four-dimensional field provided by the AIMOS model (Wissing and Kallenrode, 2009) version 1.2.

**HAMMONIA**

The Hamburg Model of the Neutral and Ionized Atmosphere (HAMMONIA) is an upward extension of the ECHAM5 atmospheric general circulation model (Roeckner et al., 2006). The model's dynamics and radiation are fully coupled to the

chemical Model of Ozone and Related Tracers (MOZART, Kinnison et al., 2007). A detailed description of the model is given by Schmidt et al. (2006). To simulate the effects of EPP, HAMMONIA is modified to incorporate the ion chemistry of the E- and F-region as described in Kieser (2011) and Meraner and Schmidt (2016). The ion chemistry treats 5 ion-electron recombinations and 12 ion-neutral reactions including 50 neutral and 6 charged ($O^+$, $O_2^+$, $N^+$, $N_2^+$, $NO^+$, $e^-$) components. Additionally, five reactions directly involving energetic particles are considered. The corresponding reaction rates are calcu-

lated using the particle induced ionization rates provided by Atmospheric Ionization Module Osnabrück (AIMOS version 1.6) (Wissing and Kallenrode, 2009). The explicit simulation of energetic particle effects on chemistry is limited to above $10^{-3}$ hPa, whereas below this altitude the production of $N(^2D)$, $N(^4S)$ and $HO_x$ is parametrised following Jackman et al. (2005a). Photochemistry includes six reactions involving radiation at wavelengths shorter than Lyman-$\alpha$. Therefore the parametrisation of Solomon and Qian (2005) and the observed 10.7 cm solar radio flux is used. Orographic gravity waves are parametrised accord-

ing to Lott and Miller (1997), while non-orographic gravity waves are parametrised according to the Doppler-spread theory from Hines (1997). A geographically uniform isotropic gravity wave source spectrum with a constant root-mean-square (rms) wave wind-speed of 0.8 m/s launched at 830 hPa is used. Additional to the homogeneous source of gravity waves, HAMMONIA considers the generation of gravity waves from tropospheric fronts following Charron and Manzini (2002). At locations where frontogenesis occurs the gravity wave spectrum is launched with an rms wave wind-speed of 2 m/s instead of 0.8 m/s. A

more detailed description of the gravity wave scheme used in HAMMONIA is given in Meraner et al. (2016). Note also that this setting of the gravity wave parameters differs from the simulation of the same winter analysed in Pedatella et al. (2014) where the waves were launched at about 650 hPa and no frontal sources were used. In this study, HAMMONIA is run with 119 vertical levels and with a triangular truncation at wavenumber 63 (T63), corresponding to a resolution of about 1.9° in latitude and longitude. Sea surface temperature and sea ice cover are taken from the Atmospheric Model Intercomparison Project 2

(AMIP2) climatology. Output is provided every 2 hours and afterwards interpolated to the satellite geolocations. The model is nudged from 850 hPa to 1 hPa with an upper and lower transition zone to the 6-hourly values of the ERA-Interim reanalysis data (Dee et al., 2011). The 'spin-up' time is one year starting on January 1, 2008.



## WACCM

For the simulations presented here, the NCAR Community Earth System Model (http://www.cesm.ucar.edu/, Hurrell et al. (2013)) is used with the Whole Atmosphere Community Climate Model as its atmospheric component (Marsh et al., 2013) (hereinafter referred to as WACCM4). The model is forced with meteorological fields from the Modern Era Retrospective

Analysis for Research and Applications (MERRA), a NASA reanalysis using the Goddard Earth Observing System Data Assimilation System Version 5 (Rienecker et al., 2011). The forcing is achieved by relaxing temperature, zonal and meridional winds and surface pressure with a time constant of 50 hours from the surface to 40 km. Above that level the forcing is reduced linearly, so that the model is free-running between 50 km and the model top at approximately 140 km ($4.5 \times 10^{-6}$ hPa). In this 'specified dynamics' version of WACCM4, there are 88 vertical levels and the horizontal resolution is $1.9°$ latitude by $2.5°$

longitude. Heating rates and photolysis are calculated using observed daily solar spectral irradiance based on the empirical model of (Lean et al., 2005) and geomagnetic activity effects in the auroral region are parametrised in terms of the Kp index (Marsh et al., 2007). The standard WACCM chemistry is described and evaluated extensively in WMO (2010). Reaction rates are from Sander et al. (2011b). For these simulations we have modified the $N+N_2$ reaction to include two additional pathways as described in Funke et al. (2008). It should be noted that both WACCM and HAMMONIA use the same chemical solver

based on the MOZART3 chemistry (Kinnison et al., 2007), include the same set of ionized species, and use the parametrised EUV ionization rates from Solomon and Qian (2005). For these simulations the latter parametrisation has been extended to include the photoionisation of $CO_2$ in the EUV. Above $5 \times 10^{-4}$ hPa ($\sim 100$ km) ionization from electrons is calculated by the WACCM parametrised aurora. It is assumed that 1.25 N atoms are produced per ion pair and divide the N atom production between ground state, $N(^4S)$, at 0.55 per ion pair and excited state, $N(^2D)$, at 0.7 per ion pair (Jackman et al., 2005b; Porter

et al., 1976). This simulation followed the "REFC1D" protocol of the Chemistry Climate Model Initiative (Eyring et al., 2013) for the specification of time-dependent greenhouse gases and ozone depleting substances. WACCM constituent and temperature profiles were saved at the model grid point and time-step (model time-step is 30 minutes) closest to each of the MIPAS observation locations. Eddy diffusion created by the dissipation of parameterized gravity waves in WACCM depends on the value assumed for the Prandtl number, Pr, which describes the ratio of the eddy momentum flux to the eddy flux of

potential temperature or chemical species. In these simulations Pr = 4, as in the study of Garcia et al. (2014).

### 3.2  Medium-top models

#### CAO-SOCOL

Since HEPPA-I (Funke et al., 2011) the CCM SOCOL (modeling tool for studies SOlar Climate Ozone Links) has been upgraded to version 3 with substantial changes related to the advection of the species. These changes and the detailed evaluation

of the new version performance were documented by Stenke et al. (2013). The CCM SOCOL v.3 consists of the MEZON chemistry transport model (Egorova et al., 2003) and MA-ECHAM5, the middle atmosphere version of the ECHAM general circulation model (Roeckner et al., 2006), with 39 vertical levels between Earth's surface and 0.01 hPa ( 80 km). Dynamical and physical processes in SOCOL are calculated every 15 minutes within the model, while full radiative and chemical calculations



are performed every two hours. Chemical constituents are transported using a flux-form semi-Lagrangian scheme (Lin and Rood, 1996), and the chemical solver is based on a Newton-Raphson iterative method taking into account 41 chemical species, 140 gas-phase reactions, 46 photolysis reactions, and 16 heterogeneous reactions. The rate constants of the gas phase and heterogeneous reactions are taken from Sander et al. (2006). The CCM SOCOL v.3 was installed in CAO (Central Aerological

Observatory, Moscow, Russian Federation) and modified to use assimilation of the meteorological fields from the ERA-I reanalysis, which is necessary to reproduce the considered SSW and ES events in January 2009. The model is nudged from 850 hPa to 1 hPa using the Jeuken et al. (1996) approach. Orographic gravity waves are parametrised according to Lott and Miller (1997). Non-orographic gravity waves are parametrised using Hines (1997) scheme implemented to ECHAM5 with a constant root-mean-square wave wind-speed of 1.0 m/s introduced at 830 hPa for all geographical locations. The daily mean

$NO_x$ mixing ratio at 0.01 hPa from MIPAS measurements (see Sec. 4) was used as the upper boundary condition at the uppermost model layer. The $NO_x$ mixing ratio was divided between NO and $NO_2$ according to their ratio in the model for any particular time step at the second layer from the model top. Model output was interpolated in time and space to the provided satellite geolocations. For the HEPPA-II experiment, the CCM SOCOL was run with T31 horizontal spectral truncation, which corresponds approximately to 3.75° × 3.75°.

**EMAC**

The ECHAM/MESSy Atmospheric Chemistry (EMAC) model is a numerical chemistry and climate simulation system that includes sub-models describing tropospheric and middle atmosphere processes and their interaction with oceans, land and human influences (Jöckel et al., 2010). It uses the second version of the Modular Earth Submodel System (MESSy2) to link multi-institutional computer codes. The core atmospheric model is the 5th generation European Centre Hamburg general cir-

culation model (ECHAM5, Roeckner et al., 2006). For the present study we applied EMAC (ECHAM5 version 5.3.02, MESSy version 2.50) in the T42L90MA-resolution, i.e., with a spherical truncation of T42 (corresponding to a quadratic Gaussian grid of approx. 2.8 by 2.8 degrees in latitude and longitude) with 90 vertical hybrid pressure levels up to 0.01 hPa. The model is nudged to ERA-Interim reanalysis data from the surface to 0.2 hPa (with decreasing nudging strength in the transition region in the five levels above) using the nudging coefficients suggested in Jeuken et al. (1996). The upper boundary condition for $NO_x$

is prescribed in the top 4 layers (0.01 hPa to 0.09 hPa) of the model. For gravity waves we used the submodel GWAVE which contains the original Hines non-orographic gravity wave routines (Hines, 1997) from ECHAM5 in a modularised structure. We tuned the parameter rmscon (root-mean-square wind-speed at bottom launch level of 642.9 hPa), which controls the dissipation of gravity waves, to 0.8 m/s. For gas phase reactions we used the submodel MECCA (Sander et al., 2011a) and for photolysis the submodel JVAL (Sander et al., 2014). 110 gas phase reactions and 44 photolysis reactions were included. Kinetical con-

stants and absorption cross-sections have mainly been taken from (Sander et al., 2011b). The $NO_x$ family was reduced to NO and $NO_2$. The chemical tracers were initialized from a multi-annual EMAC model run. Model output was done for each time step (10 minutes) which afterwards was interpolated to the satellite geolocations.



**FinROSE**

FinROSE is a global 3-dimensional CTM (further developed model version of the one described by Damski et al. (2007)). The model dynamics for the whole model domain is forced with external meteorological data, whereas the vertical wind is calculated inside the model by using the continuity equation. In this study FinROSE is nudged with ECMWF operational

analysis data. This means that changes in the atmospheric composition do not affect the model dynamics, and gravity wave parameterization is included already in the meteorological forcing data. FinROSE reproduces the distributions of 41 species from the stratosphere up to the mesosphere and lower thermosphere, and includes also about 120 homogeneous reactions and 30 photodissociation processes. Chemical kinetic data, reaction rate coefficients, and absorption cross sections are taken from look-up-tables based on the Jet Propulsion Laboratory compilation by Sander et al. (2006) and regularly updated from the

available supplements. Photodissociation frequencies are calculated using a radiative transfer model (Kylling et al., 1997). In addition to homogeneous chemistry, the model also includes heterogeneous chemistry, i.e., formation and sedimentation of polar stratospheric clouds (PSCs) and reactions on PSCs. The model is designed for middle atmospheric studies and thus the chemistry is not defined in the troposphere, but the tropospheric abundances are given as boundary conditions. For this study, the UBC for $NO_x$ (i.e. $NO + NO_2$) was implemented in the MLT region at about 0.03–0.01 hPa (the top two model layers).

The model was run with 41 vertical levels ($\sim$0–80 km) with a horizontal resolution of $6° \times 3°$ (longitude$\times$latitude). Output in the satellite geolocations was composed already during the model run by finding the closest model grid point and time step to every geolocation.

**KASIMA**

The KASIMA model is a 3D mechanistic model of the middle atmosphere including full middle atmosphere chemistry (Kouker

et al., 1999). The model can be coupled to specific meteorological situations by using analysed lower boundary conditions and nudging terms for vorticity, divergence and temperature. Here the version used for the HEPPA I experiment has been applied (Funke et al., 2011) but with a horizontal resolution of about $2.8° \times 2.8°$(T42). The model has 63 pressure levels between 7 and 120 km (chemistry up to 90 km) and a vertical resolution in the lower stratosphere of 750 m, gradually increasing to 3.8 km at the upper boundary. The frequency of output is every 6 hours. The model is nudged to ERA-Interim analyses below 1 hPa. A

numerical time step of 6 min was used in the experiments. The model uses a Lindzen-type parametrisation (Holton, 1982) to include the effect of breaking gravity waves , but no specific parametrisation of orographic gravity waves. Further details of the model are found in Funke et al. (2011). The UBC for $NO_x$ was set at the 0.3 hPa level, and not above. This occasionally causes deviations between the observations and the model above this level.

**SOCOL**

The applied version of the CCM SOCOL improves upon CAO-SOCOL, and was prepared for participation in the IGBP/SPARC CCMI project. The tropospheric chemistry component was extended by adding the Mainz Isoprene Mechanism (MIM-1), which comprises 16 organic species and a further 44 chemical reactions (Poeschl et al., 2000). The cloud influence on photolysis



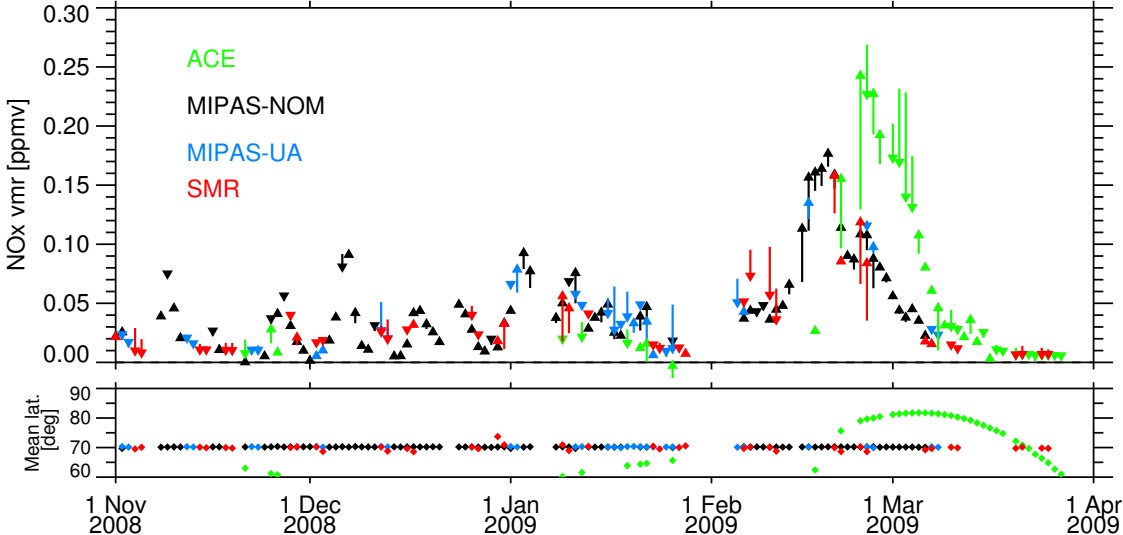

**Figure 1.** Upper panel: Daily averaged $NO_x$ mixing ratios of the upper boundary condition at 0.022 hPa and deviations from daily averaged $NO_x$ satellite observations within 60–90°N (black=MIPAS-NOM, blue = MIPAS-UA, red= SMR/Odin, green = ACE-FTS), represented by arrows (arrow head: upper boundary condition sampled at the observations' time and location, arrow origin: observation, with the arrow length representing the bias between both). Lower panel: average latitude of observations. All averages are area-weighted. Note that the arrows show up as as triangles in the case that the upper boundary condition coincides with the observations.

rates was introduced using a cloud modification factor (Chang et al., 1987). Interactive lightning source of $NO_x$ was introduced following the Price and Rind (1992) approach and adopting local scaling factors based on satellite measurements. The kinetic constants and absorption cross-sections were updated following Sander et al. (2011b). The new parametrisation of the UV heating rates (Sukhodolov et al., 2014) as well as $NO_x$ and $HO_x$ production by energetic particles (Rozanov et al., 2012) were

5   adopted. For HEPPA-II the model was run with T42 horizontal resolution, which corresponds approximately to 2.8° × 2.8°, and 39 vertical levels between the ground and 0.01 hPa. The nudging set-up and UBC for $NO_x$ are the same as in CAO-SOCOL.

## 4   $NO_x$ upper boundary condition for medium-top models

The UBC for $NO_x$ mixing ratio has been constructed from MIPAS-NOM observation data versions v4o_NO_200 and v4o_NO2_200 by projecting individual observations onto a regular grid in longitude, latitude, pressure level, and time with daily cadence us-

10   ing a distance-weighting algorithm. All observations taken within ±12 h time difference, ±10° latitude, and ±25° longitude have been considered at each grid point (weighted by the inverse distance squared) and have been vertically interpolated to a fixed pressure grid. Data gaps in space and time have been filled by linear interpolation. Note that in the model-measurement intercomparisons a newer version of MIPAS $NO_x$ is used, which was not available when the upper boundary condition was generated prior to the model runs. The horizontal resolution of the $NO_x$ UBC is 1.25°×2.5°(latitude × longitude). Thirteen





vertical pressure levels within 1–0.01 hPa are covered to allow for interpolation to the respective upper lid of the models. The $NO_x$ UBC has been evaluated by comparing with available satellite observations (see Fig. 1). To avoid sampling errors in the comparisons, the UBC field has been sampled at the measurements' locations of each day before averaging over the polar cap region. In general, there is very good agreement (within 10–20%) with independent $NO_x$ observations. However, larger

differences up to 20–50% occur sporadically for observations close to the vortex edge (e.g., when comparing to ACT-FTS at the end of February) where horizontal gradients are very pronounced.

## 5    Intercomparison strategy

The discrete horizontal sampling of satellite observations can cause large uncertainties in intercomparisons of observed and modelled averaged quantities, particularly if the sampling is sparse, irregular, or variable in time (Toohey et al., 2013). To reduce

the impact of sampling errors, we follow the same approach that was successfully applied in the first HEPPA intercomparison study (Funke et al., 2011): the model output has been sampled at the locations and times of the individual observations and has been vertically interpolated to the observed pressure levels. If available (i.e., in the case of MIPAS and MLS) , averaging kernels have been applied to the model results as described in Funke et al. (2011). Profiles have only been considered in the vertical range where the instruments' sensitivity is high enough to provide meaningful data; the remaining profile regions have

been excluded in both observations and model results.

    Model-measurement comparisons were performed on basis of daily and/or quasi-monthly averaged zonal mean data, which have been calculated in the same way for both observations and simulations. For most comparisons, data has been further binned within 70–90°N applying area-conserving $(\cos(\theta))$ weights. Note however, that the sampled portion of this latitude bin varies from instrument to instrument, making a direct comparison of the observational results difficult. However, the comparison of

model biases with respect to different observational datasets is mostly unaffected. The binning has been extended to 60–90°N in the comparisons to ACE-FTS data in order to allow for evaluations prior to February 2009. We recall that ACE-FTS has a discrete but time-varying latitude coverage (see Fig. 1) such that the resulting averages represent only a small fraction of the entire bin.

## 6    Upper mesosphere and lower thermosphere

In this section $NO_x$, CO, and temperature fields of the high-top models 3dCTM, HAMMONIA, and WACCM are compared to the observations in the MLT, the source region of odd nitrogen produced by EPP. Although, strictly speaking, temperature is not a tracer of vertical motion, the adiabatic warming during periods of strong descent introduces observable changes of the thermal structure of this region which can be used as diagnostics of vertical transport in the models. The simultaneous evaluation of modelled $NO_x$, CO, and temperature distributions allows then to attribute model biases to deficiencies in the

simulation of either particle-induced $NO_x$ production or of dynamics.



**Figure 2.** Observed and modelled $NO_x$ VMRs of MIPAS and ACE (upper two rows) and NO of SMR (lower row) in NH polar MLT region during November 2008–March 2009. Model output of the "high-top" models 3dCTM, HAMMONIA, and WACCM has been sampled at the locations and times of the observations (MPAS-UA, ACE-FTS, and SMR) for comparison. Pink lines indicate the observed VMR levels of 0.1, 1, and 10 ppmv. White regions reflect missing or not meaningful data.

Figure 2 shows the vertical distribution of NH polar $NO_x$ over time in the simulations and MIPAS-UA, ACE-FTS, and ODIN-SMR observations at 0.1 to $2 \times 10^{-4}$ hPa. SCIAMACHY observations of NO densities have not been included in this figure because NH polar observations are only available after the beginning of February. Note that MIPAS-UA and ACE-FTS provided $NO_x$ volume mixing ratios (VMR), while SMR observed NO VMR, only. This, however, introduces differences only below approximately 0.01 hPa since $NO_x$ is entirely in the form of NO above. The comparisons with the three instruments provide a consistent picture of model biases. While WACCM and HAMMONIA reproduce the observations fairly well during the whole time period in the upper mesosphere and lower thermosphere (above the 0.01 hPa level), 3dCTM exhibits too small $NO_x$ abundances in this vertical region. Below the 0.01 hPa level and during the pre-SSW phase of the winter (November–



January), WACCM and HAMMONIA agree well with the observations while 3dCTM overestimates $NO_x$ in this vertical region during most of the pre-SSW phase.

The SSW event starts with the breakdown of the polar vortex, and the dilution of the mesospheric $NO_x$ by upwelling and and increased horizontal mixing. This is clearly observed by MIPAS and SMR as a decrease of $NO_x$ between roughly 0.01 and 0.001 hPa. This initial $NO_x$ decrease is captured well by WACCM and 3dCTM, though it is too weak in the HAMMONIA simulation. The initial decrease of $NO_x$ during the SSW is followed by strong downwelling of $NO_x$ leading to a pronounced increase of mesospheric $NO_x$ and the development of the characteristic $NO_x$-"tongue". This is qualitatively captured by all models, however, the amount of $NO_x$ transported into the lower mesosphere (below 0.01 hPa) is significantly underestimated. The timing of the onset of the enhanced descent varies considerably among the models and, compared to the observations, occurs slightly too early in HAMMONIA and too late in 3dCTM. The onset of ES-related $NO_x$ increases in WACCM coincides with the observed onset, however, the modelled increases appear to last for a shorter time.

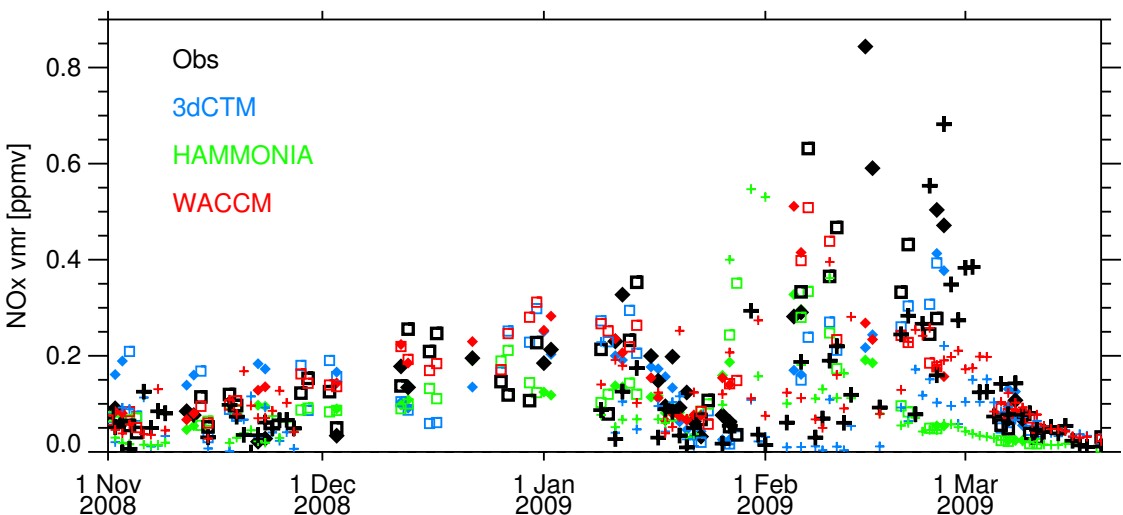

**Figure 3.** Daily averaged observed and modelled $NO_x$ at $8\times10^{-3}$ hPa within 70–90°N (60–90°N for ACE-FTS). Blue: 3dCTM, green: HAMMONIA, red: WACCM, black: observations. Model output sampled at the different instrument's locations at times is shown by different symbols: filled diamonds: MIPAS-UA, open squares: SMR/Odin, crosses: ACE-FTS.

These differences in the onset time of enhanced downwelling after the SSW are also visible in Figure 3, which shows the temporal evolution of the observed and modelled $NO_x$ at $8\times10^{-3}$ hPa. The large spread between the $NO_x$ concentrations of the same model sampled at different instruments' locations is remarkable during the post-SSW phase (February–March) and is considerably smaller before the event. This larger spread after the event is caused by the pronounced horizontal and vertical $NO_x$ gradients after development of the $NO_x$ tongue, highlighting the need to account for instrumental sampling patterns when comparing models to observations during perturbed periods.





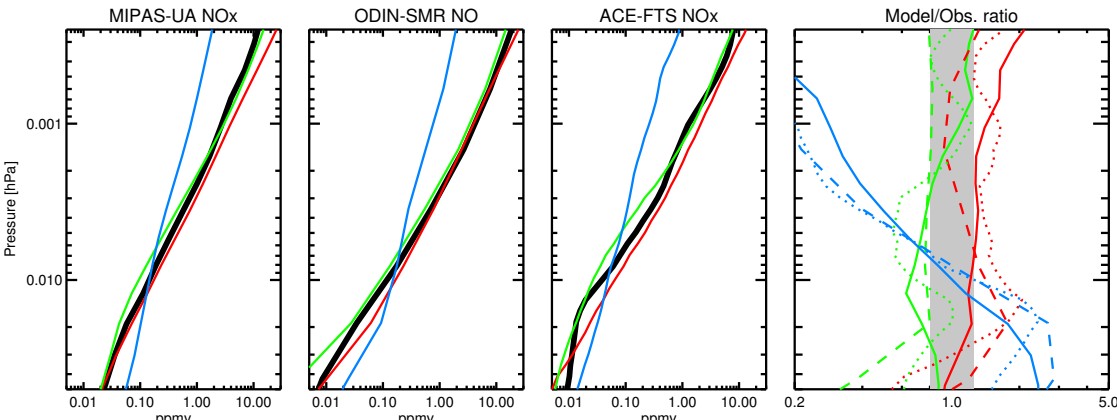

**Figure 4.** Comparison of observed polar midwinter $NO_x$ mean profiles (thick black lines) a to 3dCTM (blue), HAMMONIA (green), and WACCM (red). Right panel: ratio of model results and MIPAS-UA (solid), SMR/Odin (dashed), and ACE-FTS (dotted) observations. The grey shaded area indicates the $\pm 25\%$ range. Data have been averaged over 70–90°N and 5 December 2008 – 12 January 2009 (60–90°N and 5 November 2008 – 12 January 2009 in the case of ACE-FTS).

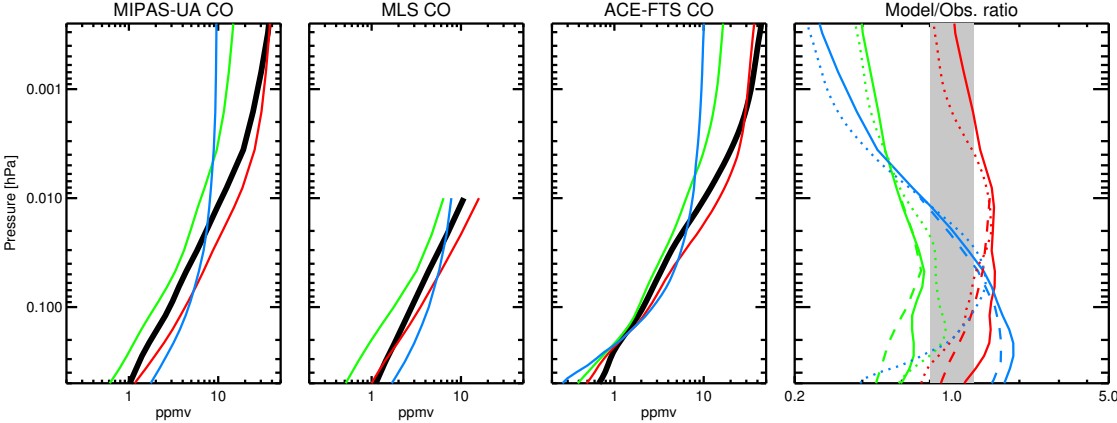

**Figure 5.** Comparison of observed polar midwinter CO mean profiles (thick black lines) to 3dCTM (blue), HAMMONIA (green), and WACCM (red). Right panel: ratio of model results and MIPAS-UA (solid), MLS (dashed), and ACE-FTS (dotted) observations. The grey shaded area indicates the $\pm 25\%$ range. Data have been averaged over 70–90°N and 5 December 2008 – 12 January 2009 (60–90°N and 5 November 2008 – 12 January 2009 in the case of ACE-FTS).

## 6.1 Unperturbed early (pre-SSW) phase

In the following, the observed and modelled vertical structure of $NO_x$, CO, and temperature during mid-winter (pre-SSW phase) is analysed in more detail to evaluate the models' ability to reproduce the EPP indirect effect for unperturbed condi-





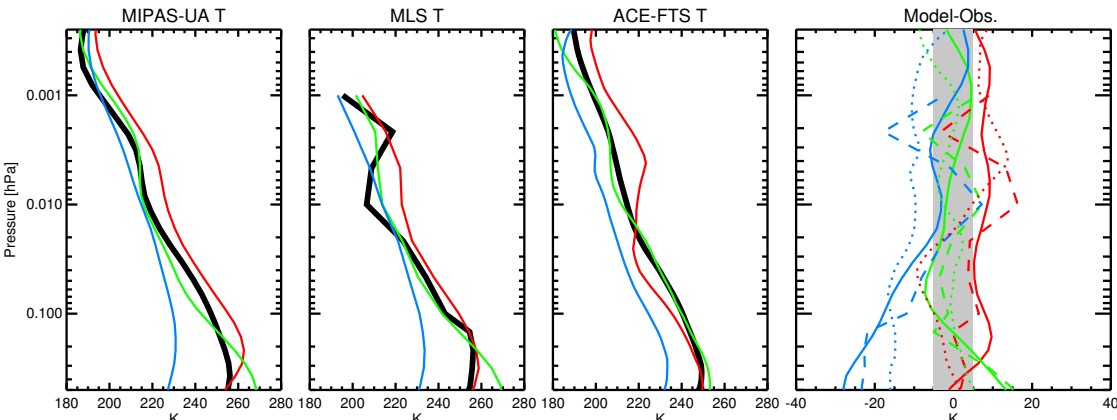

**Figure 6.** Comparison of observed polar midwinter temperature mean profiles (thick black lines) to 3dCTM (blue), HAMMONIA (green), and WACCM (red). Right panel: temperature difference of the simulations and MIPAS-UA (solid), MLS (dashed), and ACE-FTS (dotted) observations. The grey shaded area indicates the $\pm 5$ K range. Data have been averaged over 70–90°N and 5 December 2008 – 12 January 2009 (60–90°N and 5 November 2008 – 12 January 2009 in the case of ACE-FTS).

tions. Figure 4 compares the observed and modelled $NO_x$ mid-winter mean profiles averaged over 70–90°N and 5 December 2008 – 15 January 2009 (60–90°N and 5 November 2008 – 15 January 2009 in the case of ACE-FTS) above the altitude of 0.05 hPa. The observed vertical structure of $NO_x$ is reasonably well reproduced by HAMMONIA and WACCM during this period. Differences with respect to the observations are mostly within 20–50%, with WACCM being overall more on

the high side and HAMMONIA more on the low side (particularly at altitudes below 0.002 hPa). As discussed earlier, the 3dCTM simulations show a much less pronounced vertical gradient resulting in a significant (in terms of the observational spread) $NO_x$ underestimation (up to a factor of 8) at altitudes above $10^{-2}$ hPa and overestimation (up to a factor of 3) below. Figure 5 compares the corresponding mean profiles of CO, observed by MIPAS-UA, MLS, and ACE-FTS above the altitude of 0.5 hPa. Again, WACCM and HAMMONIA show a vertical gradient that is roughly in agreement with the observations.

On the other hand, the absolute CO values of WACCM are slightly (up to 40%) higher while HAMMONIA underestimates the CO abundances by a factor of 2—3. The latter can be explained by missing thermospheric production mechanisms in the model, specifically the $CO_2$ photolysis in the extreme ultraviolet (at wavelengths < 121 nm) and the reaction of $CO_2$ with the atomic oxygen ion (Garcia et al., 2014), that act in addition to the photolysis of $CO_2$ in Lyman-alpha and the Schuman-Runge continuum. The 3dCTM simulations, similarly as for $NO_x$, show a too weak gradient in the mesosphere compared to the obser-

vations, resulting in an underestimation above 0.03 hPa and an overestimation below. The corresponding temperature profiles (see Figure 6), observed by MIPAS-UA, MLS, and ACE-FTS (note that SABER is not included because the observations in December cover only up to 52°N) indicate good agreement with the observations for HAMMONIA and a slight warm bias of 5–10 K for WACCM. Mesospheric 3dCTM temperatures are systematically too cold by 10–30 K in the middle and lower mesosphere.



The good overall agreement of $NO_x$, CO, and temperature from HAMMONIA and WACCM with the observations in December suggests that both $NO_x$ sources and dynamical conditions are well represented by these models, allowing for an adequate description of the EPP indirect effect in the MLT during unperturbed conditions early in NH winters. Interestingly, the consideration of mid-energy electron (MEE)-induced ionization in HAMMONIA (via AIMOS) does not introduce noticeable

differences in the NO distribution with respect to WACCM, the latter only accounting for auroral electrons. This suggests that the impact of MEE during the solar minimum 2008/2009 NH winter was rather small. 3dCTM simulations, on the other hand, show significant discrepancies with the observations. The similarity of the model bias in the vertical gradients of $NO_x$ and CO suggests that these differences with respect to the observations are due to the representation of dynamics in 3dCTM rather than to the EPP source. The vertical gradient of the 3dCTM CO and $NO_x$ profiles both show too low values in the

lower thermosphere, and too high values in the upper to mid mesosphere. The underestimation of lower thermospheric CO is likely due to the model chemistry as, like in HAMMONIA, neither the EUV photolysis of $CO_2$ nor the production of CO by positive ion chemistry in the lower thermosphere are considered in 3dCTM. The underestimation of thermospheric $NO_x$ could be caused by a too weak NO production or too fast transport out of the (polar) source region, either by horizontal mixing, or across the mesopause. The high values of both CO and $NO_x$ in the mesosphere on the other hand are likely due to the

representation of mesospheric dynamics in 3dCTM, which is driven by temperatures and wind fields from the LIMA model. A likely reason seems the neglect of sub-scale ($\leq 500$ km) gravity waves in the LIMA model leading to an underestimation of the GW drag throughout the mesosphere, but to an overestimation in the lowermost thermosphere (see Section 2.7). This leads to a suppression of vertical motion in the mesosphere which is also reflected in a negative bias in temperatures, and consequently, to an accumulation of CO and $NO_x$.

**6.2   Perturbed late (post-SSW) phase**

Figure 7 compares the observed and modelled $NO_x$ February mean profiles corresponding to the perturbed post-SSW phase of this winter, characterized by enhanced descent of $NO_x$. This comparison includes also SCIAMACHY NO density averages. Above 0.005 hPa, a larger spread of model-measurement differences compared to December is found, likely related to the enhanced spatial and temporal variability (see also Fig. 3). On average, however, these differences are very similar to

those encountered during mid-winter. Below 0.005 hPa, all models systematically underestimate the observed $NO_x$ increases associated with the ES event by a factor of 2–3.

Adiabatic heating associated with the enhanced mesospheric descent is responsible for the reformation of the stratopause at a pressure level as high as 0.005 hPa. Figure 8 shows the temporal evolution of the vertical temperature structure at 70–90°N in January–March as observed by SABER and simulated by 3dCTM (LIMA), HAMMONIA, and WACCM. We have chosen this

observational dataset for the comparison to the models because of its full temporal coverage in this period and the high vertical resolution in the entire vertical range. The observed elevated stratopause started to develop at the beginning of February and remained at around 0.01 hPa for a month before it descended to its climatological height in the course of March. The highest stratopause temperatures during the elevated phase were reached around 20 February. Although all models simulate an elevated stratopause, its temporal evolution differs significantly from the observations. HAMMONIA and WACCM show an ES onset




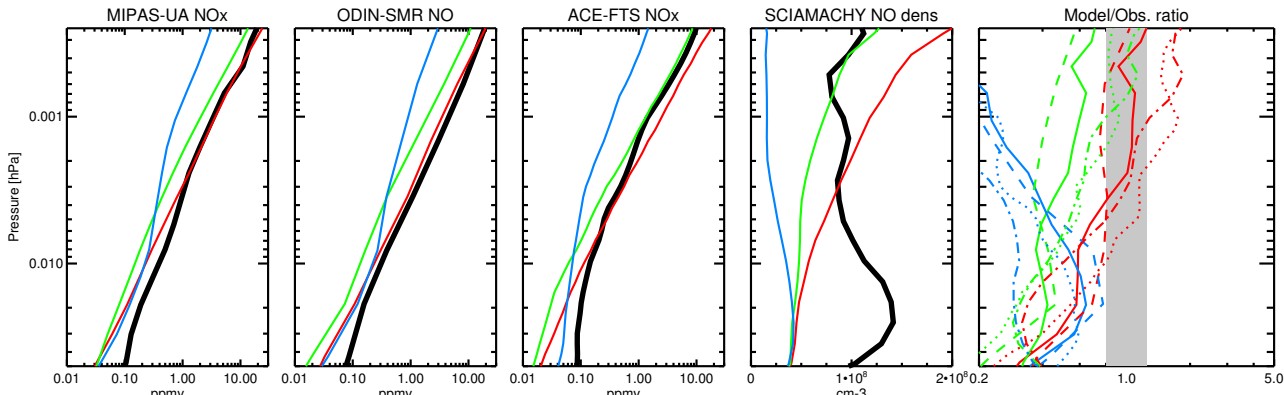

**Figure 7.** Comparison of observed $NO_x$ mean profiles (thick black lines) for February 2009 (during the ES event) and 70–90°N to 3dCTM (blue), HAMMONIA (green), and WACCM (red). Right panel: ratio of model results and MIPAS-UA (solid), SMR-Odin (dashed), ACE-FTS (dotted) and SCIAMACHY (dash-dotted) observations. The grey shaded area indicates the ±25% range. Data have been averaged over 70–90°N and 1 February 2009 – 1 March 2009 (60–90°N and 1 February 2008 – 15 March 2009 in the case of ACE-FTS).

and formation level similar to the observed ones, however, highest temperatures at this level are reached immediately after the onset, about 20 days earlier than in the observations. In both models, the ES level starts to descend immediately after its formation, more quickly than observed, and faster in HAMMONIA than in WACCM. During the descent, the modelled stratopauses become increasingly warmer. 3dCTM, in contrast, simulates a much later onset (about 2 weeks after the observed

one) and the ES temperatures are much colder than in the observations. However, the modelled ES remains at an elevated level for a longer time (although slightly lower than the observed ES) and the time delay until reaching the maximum ES temperatures is comparable to the observed temperature evolution. These differences between 3dCTM on the one hand, and WACCM, HAMMONIA, and mostly also the observations, on the other hand, highlight the role of subscale gravity waves for the temporal evolution of the ES event. The onset of the SSW event is driven mainly by large-scale planetary waves breaking

down the horizontal circulation, and is captured comparatively well by all three models. However, the reformation of the stratopause at upper mesospheric altitudes is driven by small-scale gravity waves reaching up to the upper mesosphere after the event. As these smaller gravity waves are essentially missing in the LIMA data, the build-up of the elevated stratopause is delayed in 3dCTM, and its strength is weaker.

     To investigate whether the encountered differences between the models and SABER data are robust with respect to instru-

mental uncertainties, we extend the analysis to MIPAS-UA, ACE, and MLS temperature observations and compare the model differences to all observations (see Figure 9). Despite minor changes related to the different latitude range covered by the instruments, the encountered model biases are consistent for all instruments, indicating a too cold mesosphere of 3dCTM, and a dipole type pattern in HAMMONIA and, less pronounced, in WACCM with colder temperatures after the ES onset in the upper mesosphere and warmer temperatures below.





**Figure 8.** Temporal evolution of daily averaged polar cap temperatures at 4–0.0005 hPa from SABER observations and simulations of 3dCTM, HAMMONIA, and WACCM (from top left to bottom right). The white contours correspond to the observed temperatures of 220 and 240 K.

A similar analysis of NH polar temperature evolution in early 2009 in several whole atmosphere models (including HAMMONIA) and MLS observations has been performed by Pedatella et al. (2014). Their Figure 1 can be directly compared to our Fig. 8. In agreement with our results, most of the investigated models in the study of Pedatella et al. (2014) did not maintain the stratopause height near 0.01 hPa until the end of February as in the observations, except WACCM-X, which was nudged to NOGAPS-ALPHA reanalysis data (assimilating observed temperatures) up to 92 km. Siskind et al. (2015) further showed with WACCM simulations of the same NH winter, that nudging to a more realistic meteorology (with an ES evolution closer to the observations) up to 92 km dramatically improves the simulated NO descent during this event compared to SOFIE observations.

Unresolved non-orographic GWD is thought to play a crucial role in the strengthening of mesospheric descent in the vicinity of the NO source region during ES events by providing enhanced westward momentum which forces a poleward and downward



**Figure 9.** Top: temporal evolution of daily averaged polar cap temperatures at 4–0.0005 hPa observed by MIPAS-UA, MLS/Aura, ACE-FTS, and SABER (from left to right). Bottom: Corresponding differences between temperatures simulated with the "high-top" models (3dCTM, HAMMONIA, and WACCM) and the observations.

residual circulation (McLandress et al., 2013; Siskind et al., 2015). Motivated by the results of our analysis, Meraner et al. (2016) investigated the sensitivity of the HAMMONIA model to changes in the parametrisation of non-orographic gravity waves. By weakening the amplitude of the gravity waves at the source level, they could substantially improve the modelled temperature and $NO_x$ increases (both in terms of timing and amount) compared to the MIPAS observations. They found that the amount of transported $NO_x$ depends strongly on the altitude at which momentum is deposited in the mesosphere. Smaller gravity wave amplitudes favour the wave breaking and momentum deposition at higher altitudes, closer to the NO source region.



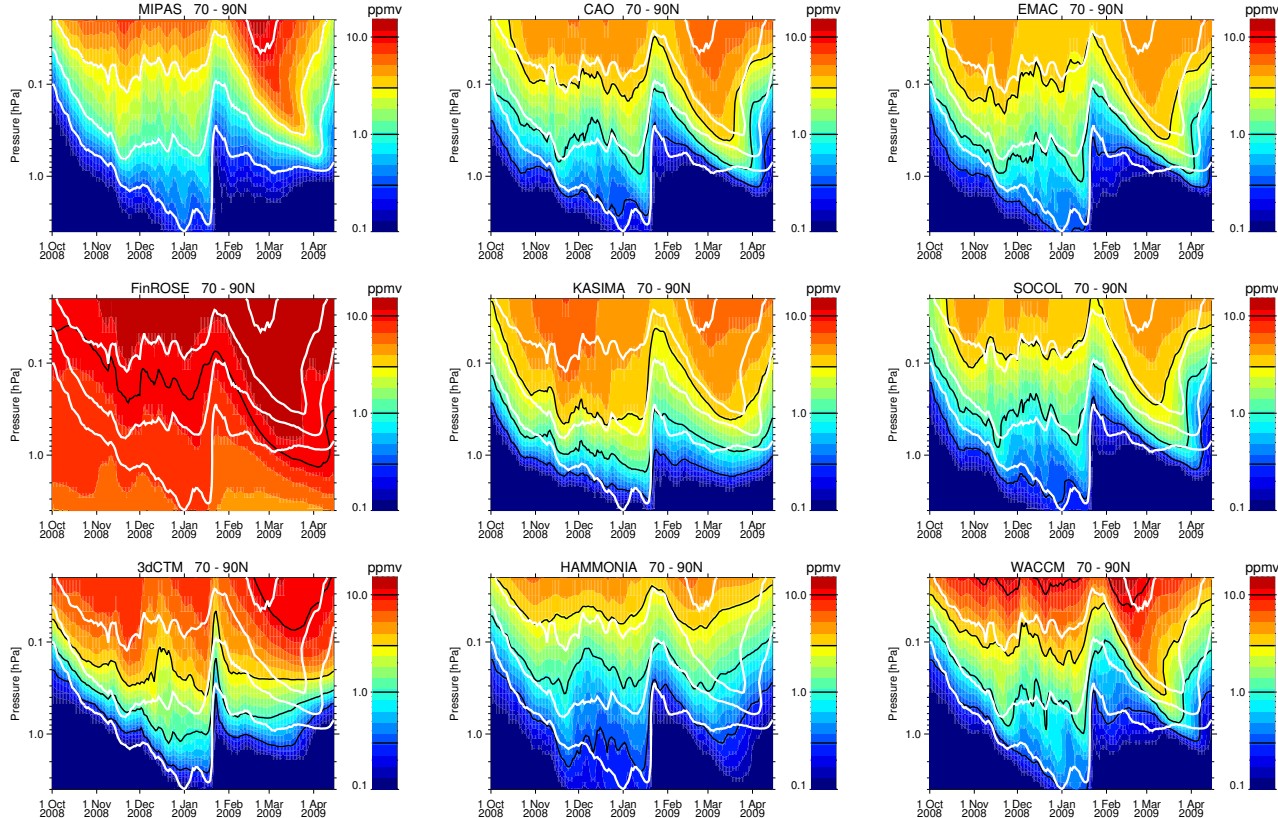

**Figure 10.** MIPAS-NOM and modeled temporal evolutions of CO at 4–0.02 hPa within 70–90°N. White lines indicate the observed VMR levels of 0.3, 1, 3, and 10 ppmv.

The structural similarities of HAMMONIA and WACCM temperature biases suggest that changes in the non-orographic GWD parametrisation might also improve the representation of $NO_x$ descent during ES events in WACCM.

# 7 Upper stratosphere and mesosphere

In this section CO, $NO_x$, and temperature fields of all involved models are compared to the observations in the upper stratosphere and mesosphere (USM). The aim is to evaluate the models' ability to reproduce $NO_x$ transport into the stratosphere during both the unperturbed pre-SSW phase and the ES event, and to identify whether discrepancies with respect to the observations are related to dynamics or chemistry. The latter is of particular concern for the medium-top models applying the $NO_x$ upper boundary condition.





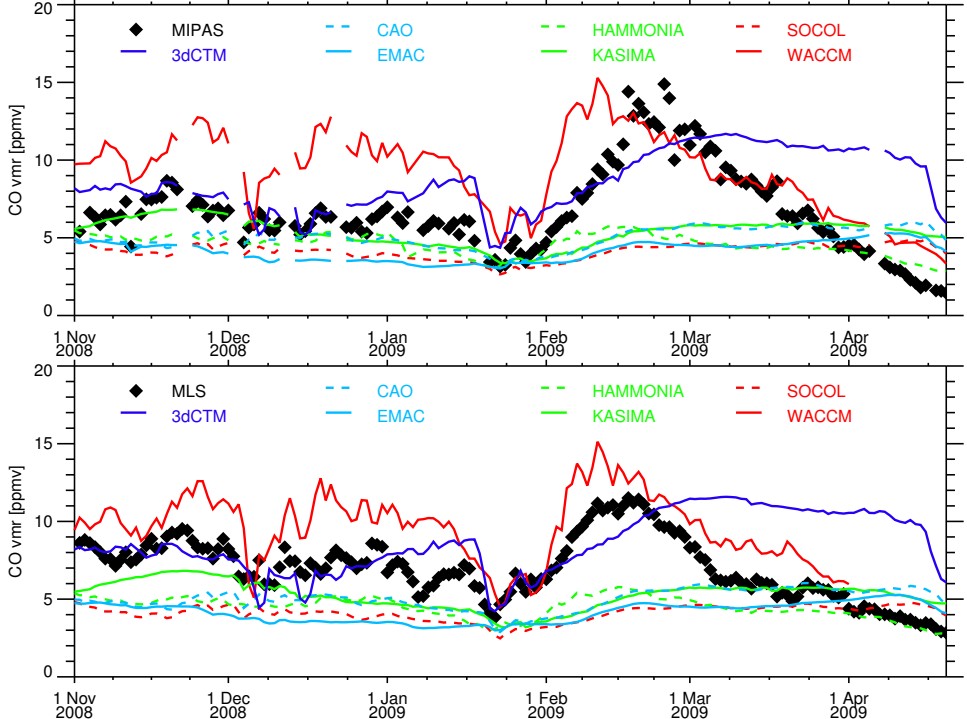

**Figure 11.** MIPAS-NOM (top) and MLS/Aura (bottom) temporal evolutions of CO VMR in comparison with the model results within 70–90°N at 0.02 hPa.

### 7.1 CO

CO is an excellent tracer of vertical motion in the USM during polar winter because of its pronounced vertical gradient in this region and the long chemical lifetime under dark conditions. Further, the relatively less pronounced gradient at higher altitudes (compared to $NO_x$) results in a weaker sensitivity to dynamical variability in the MLT, hence allowing to study the descent in the USM separately. In addition, the very low stratospheric CO background concentrations allow to trace mesospheric descent down to altitudes below 30 km without the need to invoke tracer correlations as in the case of odd nitrogen (Funke et al., 2014a).

CO observations are available from MLS, ACE, and MIPAS. As an example, Figure 10 compares the MIPAS-NOM CO temporal evolution with the models. At a first glance, the observed evolution of the CO vertical distribution is qualitatively well reproduced by most models, except for FinROSE which exhibits a very weak vertical gradient all over the winter. This behaviour is caused by a simplified $CO_2$ representation leading to overestimation of CO production and a largely enhanced CO background in the middle and upper atmosphere. All other models capture the observed polar winter descent down to pressure levels around 3 hPa in the first part of the winter, the sudden reduction of CO during the SSW caused by meridional mixing and upwelling, as well as the enhanced descent during the ES event.





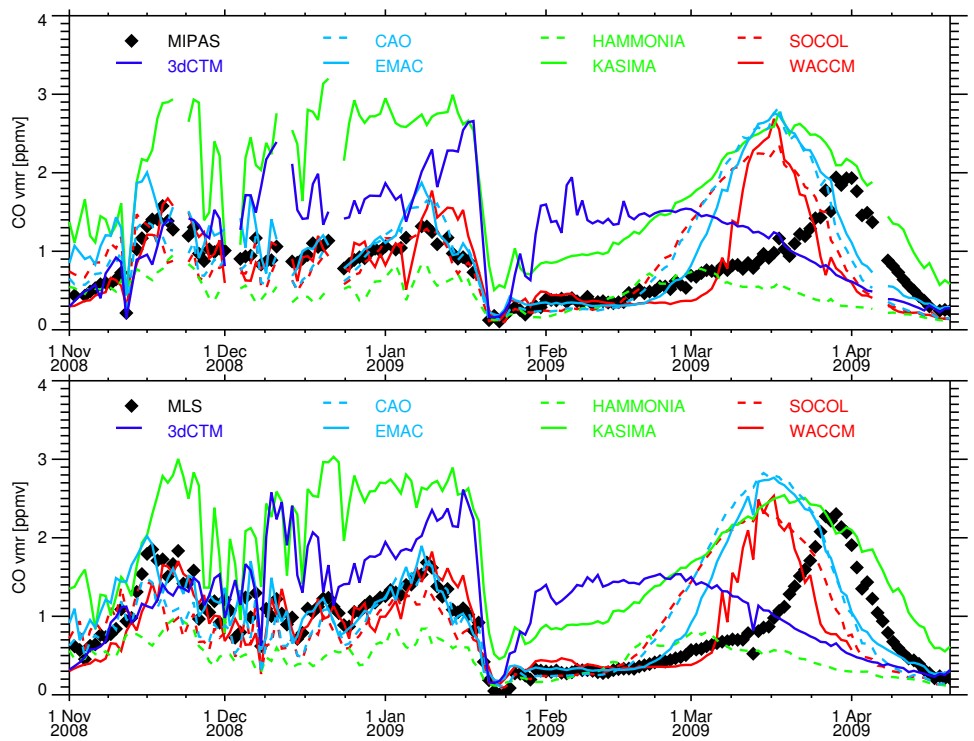

**Figure 12.** MIPAS-NOM (top) and MLS/Aura (bottom) temporal evolutions of CO VMR in comparison with the model results within 70–90°N at 0.5 hPa.

A more quantitative analysis is provided by Figures 11 and 12, comparing the modelled CO evolutions at 0.02 and 0.5 hPa, respectively, to MIPAS-NOM and MLS observations (note that FinROSE is not included here because of the unrealistically high mixing ratios). The comparisons to both instruments provide a very similar picture, hence confirming the robustness of the encountered model biases. Observed CO abundances at 0.02 hPa are around 6–8 ppmv during the pre-SSW phase, decrease to

5  4 ppmv during the SSW, and show a pronounced peak of 12–14 ppmv in February related to the ES event. Medium-top models exhibit slightly lower CO abundances (around 5 ppmv) that do not vary significantly over the winter. This behaviour is expected since transport of lower thermospheric CO into the model domain is typically not considered and, as consequence, dynamically induced variations are mostly absent at this pressure level close to the models' upper lid. As an exception, tracers are transported in KASIMA above the chemical domain at 90 km which causes accumulation effects, resulting in slightly increased abundances

10  during early winter. Further, minor differences in the late winter abundances simulated by KASIMA and CAO on the one hand and EMAC and SOCOL on other hand can be attributed to the use of different kinetic data in the chemistry schemes, primarily affecting OH involved in the CO loss reaction. The observed CO evolution at 0.02 hPa is qualitatively well captured by WACCM, although the abundances during the pre-SSW phase of about 10 ppmv are overestimated by ∼40% compared to the observations and the ES-related peak occurs earlier than in the observations. HAMMONIA CO abundances are underestimated





due to missing thermospheric CO production mechanisms (see previous section) and are very close to the CO amount simulated by the medium-top models (∼5 ppmv). 3dCTM simulates early winter CO abundances that are roughly in agreement with the observations. ES-related CO enhancements in the post-SSW phase, however, are delayed and persist for a longer period than observed.

5     The observed CO evolution at 0.5 hPa is well reproduced by most medium-top models and WACCM in the pre-SSW phase. KASIMA, and 3dCTM overestimate the CO abundances by a factor of ∼2.5 and ∼1.5, respectively, while HAMMONIA simulates about 50% lower than observed CO abundances. The ES-related CO increases peak in most models too early (around mid March) compared to the observed peak occurrence around 1 April, although the peak magnitude is reasonably well simulated (with exception of HAMMONIA). The CO peak in HAMMONIA occurs even 2 weeks earlier than in the other models. In 10  3dCTM, the CO tongue does not reach the 0.5 hPa level (see Fig. 10), likely because of the too-late formation of the elevated stratopause discussed in the previous section. The high CO abundances of this model in February, immediately after the SSW, seem to be caused by horizontal mixing, after a short period of localised upwelling during the sudden warming.

    The individual impacts of orographic and non-orographic gravity wave drag on the mesospheric CO evolution in the CMAM model has been evaluated by comparing with the same MLS observations during the 2008-2009 NH winter by McLandress 15  et al. (2013). Our Figure 12 can be qualitatively compared to their Figure 8 (although the latter shows the CO evolution at a slightly higher pressure level). The CO evolution in the CMAM simulation, including all gravity wave sources, is very similar to that obtained by most of the models included in our study (note that the apparently smaller time lag of the ES-related peak in the McLandress et al. (2013) study is related to the higher pressure level of their comparison). On the other hand, there are similarities between their simulation without orographic GWD and the KASIMA simulation presented here, 20  particularly regarding the CO overestimation in the pre-SSW phase and the relatively broad CO peak after the ES event. Note, that KASIMA does not employ a specific parameterization for orographic gravity wave drag which may be justified as KASIMA is nudged up to 1 hPa but seems not to be sufficient near the stratopause. This is also seen in the low bias of the stratopause temperature in the pre-SSW phase (see Fig. 19). Further, our 3dCTM results share some characteristics of the CMAM simulation without any GWD. In particular, both simulations exhibit a steady (though fluctuating) increase of CO until 25  the SSW, a short recovery time after the warming, and the absence of an ES-related peak in March/April. This again highlights the importance of the proportion of the gravity wave spectrum not considered in the LIMA model – the sub-scale (≤ 500 km) waves for the mesospheric meridional wintertime circulation, in particular during the recovery phase of the elevated stratopause event as discussed in the previous section, but also for the "undisturbed" pre-event period.

### 7.2  NO$_x$ in the early (pre-SSW) phase

30  In the following, the observed and modelled vertical structure of NO$_x$ in the USM during mid-winter (pre-SSW phase) is analysed in more detail to evaluate how well the models reproduce the EPP indirect effect in this region for unperturbed conditions. Figure 13 compares the NO$_x$ evolution of all models at 1–0.02 hPa with the MIPAS data. All models capture the observed early winter NO$_x$ descent characterized by a quasi-continuous increase of NO$_x$ until the SSW-related disruption in mid-January. The magnitude of the observed NO$_x$ enhancements is well reproduced by EMAC, FinROSE, KASIMA, HAMMONIA, and





**Figure 13.** MIPAS-NOM and modelled temporal evolutions of $NO_x$ in the pre-SSW phase of the 2008/09 NH winter at 1–0.02 hPa within 70–90°N. White lines indicate the observed VMR levels of 10, 20, 50, 70, 100, and 150 ppbv. White regions reflect missing or not meaningful data.

WACCM. Descending $NO_x$ can be distinguished from the background in these simulations and in the observations down to pressure levels of 0.3–0.5 hPa.





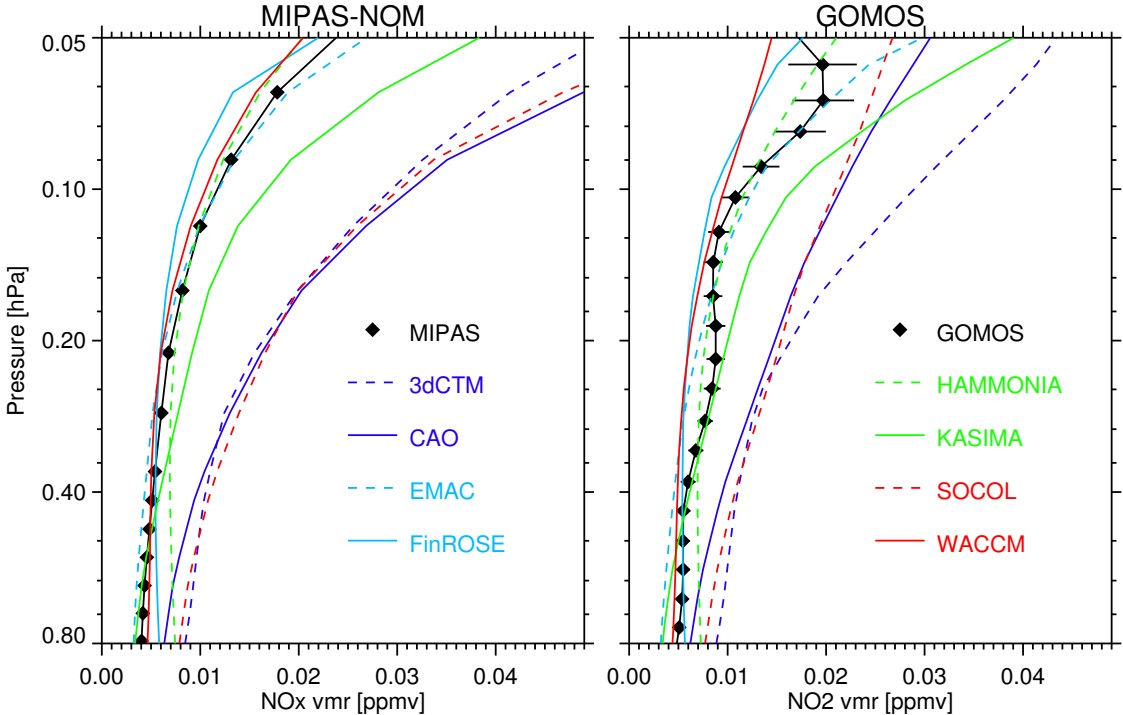

**Figure 14.** Left: MIPAS-NOM and modelled mean $NO_x$ profile for the period 15 December 2008–12 January 2009 within 70–90°N. Right: GOMOS and modelled mean nighttime $NO_2$ for the same period within 75–85°N. The error bars indicate random retrieval errors of the averaged observational data.

As discussed in Sec. 6, 3dCTM overestimates the observed $NO_x$ increasingly towards lower altitudes and shows a double peak structure (with a $NO_x$ depletion around mid-December) that is not seen in the MIPAS $NO_x$ data, though a similar feature is also observed in 3dCTM CO, and at least indicated in MIPAS CO, at the same time. Also SOCOL and CAO overestimate substantially the descending $NO_x$ amounts. Since the CO descent is well described by the latter two models, the $NO_x$ overestimation is likely related to the prescription of $NO_x$ at the upper model lid. The $NO_x$ abundances at the upper model level (0.01 hPa) are in agreement with the values specified by the UBC. However, in contrast to the observations and other models, which show a rapid decrease towards lower altitudes, the abundances remain nearly constant in the entire vertical range above 0.03 hPa. This behaviour is caused by a model boundary artefact introducing unrealistically fast vertical propagation of the $NO_x$ caused either by too high vertical velocities at the model lid or low vertical model resolution. Indeed, the descending $NO_x$ amounts are substantially reduced in a test simulation with $NO_x$ prescribed at the second layer from the top (not shown) making the SOCOL results similar to those of EMAC.

A more quantitative view of the modelled midwinter $NO_x$ profiles in comparison with observations of the MIPAS and GOMOS instruments (the latter measuring nighttttime $NO_2$) is provided in Figure 14. Other instruments measuring $NO_x$ species could not be included in this comparison: SMR because they measured only NO but most of $NO_x$ is in the form





of $NO_2$ below 0.1 hPa in dark conditions, SCIAMACHY because it is not sensitive to NO below ∼65 km, and ACE-FTS because it did not sample latitudes polewards of 70°N in midwinter. Both MIPAS and GOMOS consistently show VMRs of about 20 ppbv at 0.05 hPa, decreasing to the background values of 5 ppbv at 0.8 hPa. The observed profile is reproduced within 20% by EMAC, FinROSE, HAMMONIA, and WACCM. The KASIMA results are about 50% higher than the observations.

3dCTM, CAO, and SOCOL overestimate the observations by a factor of 2–3.

Overall, most atmospheric models are capable of providing a realistic and consistent picture of $NO_x$ descent in dynamically and geomagnetically unperturbed NH early winters as in 2008/2009. This is the case for high-top models explicitly considering odd nitrogen production by EPP in the MLT region, as well as for medium top models employing a $NO_x$ upper boundary condition. However, some individual models show significant biases in the simulated early winter $NO_x$ descent which could

be traced back to deficiencies in either the dynamical or chemical schemes.

### 7.3  $NO_x$ in the perturbed late (post-SSW) phase

Limitations of high-top models to reproduce quantitatively the observed $NO_x$ descent from the upper mesosphere during the perturbed part of the 2008/09 NH winter (post-SSW phase) have already been discussed in Sec. 6. An important question is whether medium-top models, prescribing realistic $NO_x$ distributions at the model's upper lid, could provide a better descrip-

tion of ES-induced odd nitrogen transport by bypassing the problem of underestimated descent in the region above 80 km, as encountered in the high-top models. Figures 15 and 16 show the temporal evolutions of modelled $NO_x$ during the ES event in comparison with MIPAS-NOM and ACE-FTS observations, respectively. Despite the sampling-related differences, both instruments provide a very consistent picture of model biases. In particular, the time shift (earlier occurrence) of the modelled $NO_x$ tongue (except 3dCTM), also identified in the CO comparisons, is clearly visible in the comparisons with both instruments.

Again, SOCOL and CAO overestimate significantly the observed $NO_x$ (about a factor of 5) in the descending tongue (for the reasons already identified in the midwinter comparisons). This overestimation is even more pronounced than in the pre-SSW phase. In the case of HAMMONIA, related to the fast downward propagation of the ES (see Sec. 6), the $NO_x$ peak occurs earlier and the tongue descends faster, merging with the background already in mid-February. In 3dCTM, the $NO_x$ tongue reaches the lower mesosphere (0.02 hPa) later than in the other models and in observations due to the too slow descent rates

throughout the mesosphere. Thus, the development of the $NO_x$ tongue in the lower mesosphere is delayed, and it does not reach to stratospheric altitudes.

The $NO_x$ tongue observed by MIPAS reaches the 1 hPa level by the end of April. The reversal of the residual circulation in spring disabled further downward propagation of the tongue. ACE-FTS observed polar latitudes until 25 March, when the tongue reached the 0.3 hPa level in agreement with MIPAS observations at the same time. Compared to the observations, the

$NO_x$ tongue in the model simulations (except HAMMONIA and 3dCTM) penetrates deeper, reaching the 2–3 hPa pressure levels at the end of April.

Figure 17 shows more quantitatively the observed and modelled occurrence time and magnitude of the $NO_x$ peak as a function of pressure level. The similar peak timing simulated by all models (except 3dCTM and HAMMONIA), about 2 weeks earlier than the observed peak below the 0.2 hPa level, is surprising. In the WACCM simulation, this time shift with respect to





**Figure 15.** MIPAS-NOM and modelled temporal evolutions of $NO_x$ during the ES event at 1–0.02 hPa within 70–90°N. White lines indicate the observed VMR levels of 10, 20, 50, 70, 100, 150, and 200 ppbv.

the observations is present over the whole vertical range. Interestingly, the peak occurrence time in the medium-top models, all prescribing the observed $NO_x$ evolution at their upper lid, converges with the descent to the same occurrence time as simulated by WACCM at lower altitudes, i.e. earlier than in the observations. It is worth noting that a HAMMONIA simulation (not

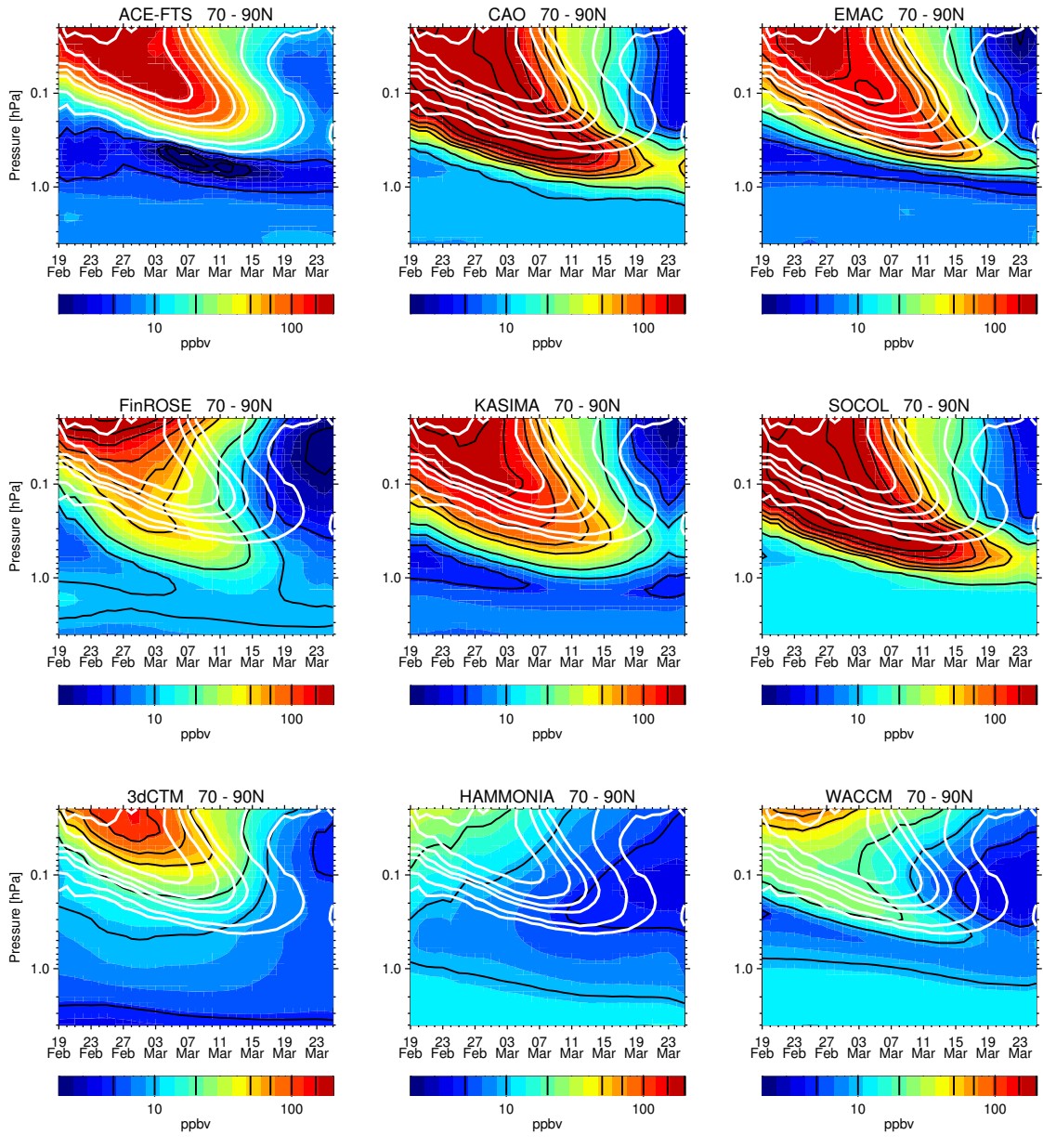

**Figure 16.** Same as Fig. 15, but for ACE-FTS.

shown) with reduced non-orographic gravity wave amplitude (Meraner et al., 2016) exhibits both a $NO_x$ peak occurrence time and magnitude in very good agreement with the observations down to pressure levels around 0.3 hPa. Below, however, the peak occurrence time in this particular HAMMONIA simulation converges again to that of most of the other simulations.



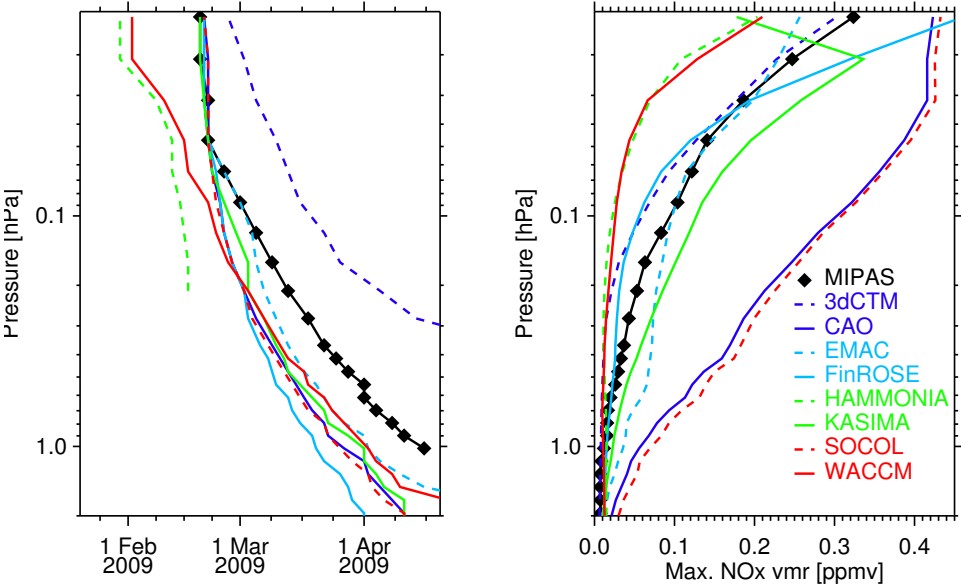

**Figure 17.** Left: MIPAS-NOM and modelled time evolution of the occurrence of the $NO_x$ peak as function of pressure after the ES event. Right: observed and modelled $NO_x$ peak values, averaged over 70–90°N.

Despite the consistency of the models with respect to the timing of the $NO_x$ descent in the lower mesosphere, indicating similar dynamical representations, the spread of the magnitude of the modelled $NO_x$ peaks (right panel of Fig. 17) is very large (within 0.2–3 times the observed magnitude), even when excluding the CAO and SOCOL results. This is particularly surprising in the case of the medium-top models, all of them prescribing the same $NO_x$ obtained from observations, and will

be discussed in more detail at the end of this section.

Figure 18 shows the temporal evolution of the MIPAS observations and modelled $NO_x$ at 0.5 hPa together with the temperature evolution slightly above, at 0.2 hPa. There is a clear link between the earlier occurrence of the modelled $NO_x$ peaks and the time shift of the modelled temperature increases after the SSW, occurring systematically about 2 weeks earlier than in the observations (with the exceptions of HAMMONIA and 3dCTM). In order to check if the temperature bias of the simulations

with respect to MIPAS is consistent with the other measurements, we show in Fig. 19 the vertical structure of the temperature differences between the medium-top models and MIPAS-NOM, MLS, ACE-FTS, and SABER observations, similarly as done for the high-top models in Sec. 6. All medium-top models show a warm bias of 15–25 K around 0.2 hPa in February and early March, and a cold bias of 5–10 K around 1 hPa during the same period (though slightly less pronounced in KASIMA). Similar biases have been detected in the WACCM simulations (see Fig. 9).

The systematic, dipole-type temperature bias of the high-top model WACCM and all medium-top models, with similar amplitudes and time evolutions, explains the consistently too early occurrence of the $NO_x$ descent encountered in these models. It also hints at a common origin. One plausible reason for the temperature bias could be the meteorological data nudged in



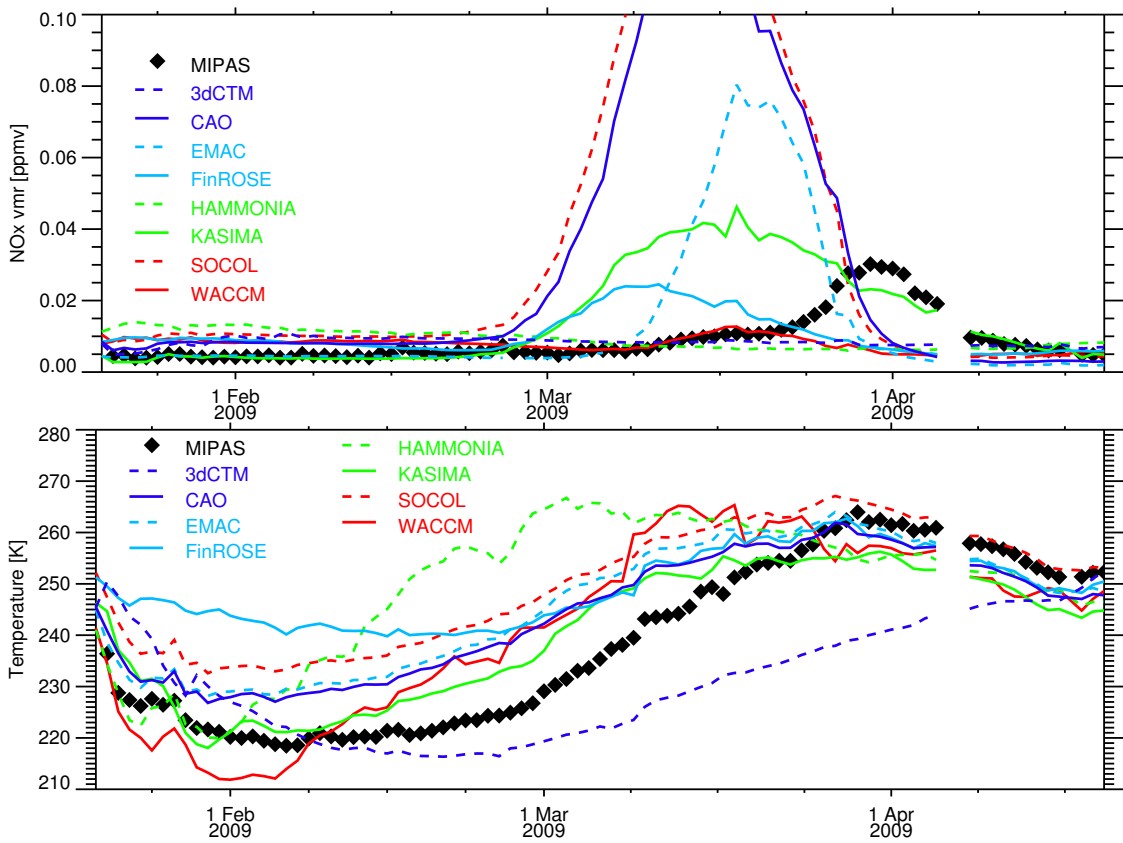

**Figure 18.** MIPAS-NOM and modelled temporal evolutions of $NO_x$ at 0.5 hPa (top) and of temperature at 0.2 hPa (bottom) within 70–90°N during the ES event.

most models below 1 hPa. Around this pressure level, a cold bias of these models is observed, including FinROSE which relies

entirely on ECMWF operational analysis data, and EMAC which applies the nudging to ERA Interim reanalysis data up to the

altitude of 0.2 hPa. This indicates that the cold bias is present already in the ECMWF operational analysis and ERA Interim

data. This bias might then likely influence the model dynamics extending above the nudged region. The cold bias around 1 hPa

5  in February is also seen in the WACCM simulation (c.f. Fig. 8), suggesting that it is also present in the MERRA reanalysis.

This is confirmed by comparison of MERRA and MLS temperatures (not shown). Only in the HAMMONIA simulation, which

shows a pronounced warm bias in the entire 2–0.1 hPa region, the local influence of the nudged meteorology at the edge of the

nudging region seems to be outweighed by the internal model dynamics. It is beyond the scope of this paper to investigate in

detail the possible mechanisms for the vertical propagation of dynamical biases, introduced by the nudging, resulting in a too

10  early descent of mesospheric $NO_x$. However, since the encountered cold bias at 1 hPa is restricted to latitudes northward of 60°

(see Fig. 20) and hence implies a strengthening of the meridional temperature gradient, it is likely to accelerate zonal winds at

this level and above, which in turn would lead to changed filtering conditions for the propagation of gravity waves. Another



**Figure 19.** Top: temporal evolution of daily averaged polar cap temperatures at 4–0.02 hPa observed by MIPAS-NOM, MLS/Aura, ACE-FTS, and SABER (from left to right). Bottom: Corresponding differences between temperatures simulated with the "medium-top" models (CAO, EMAC, FinROSE, KASIMA, and SOCOL) and the observations.





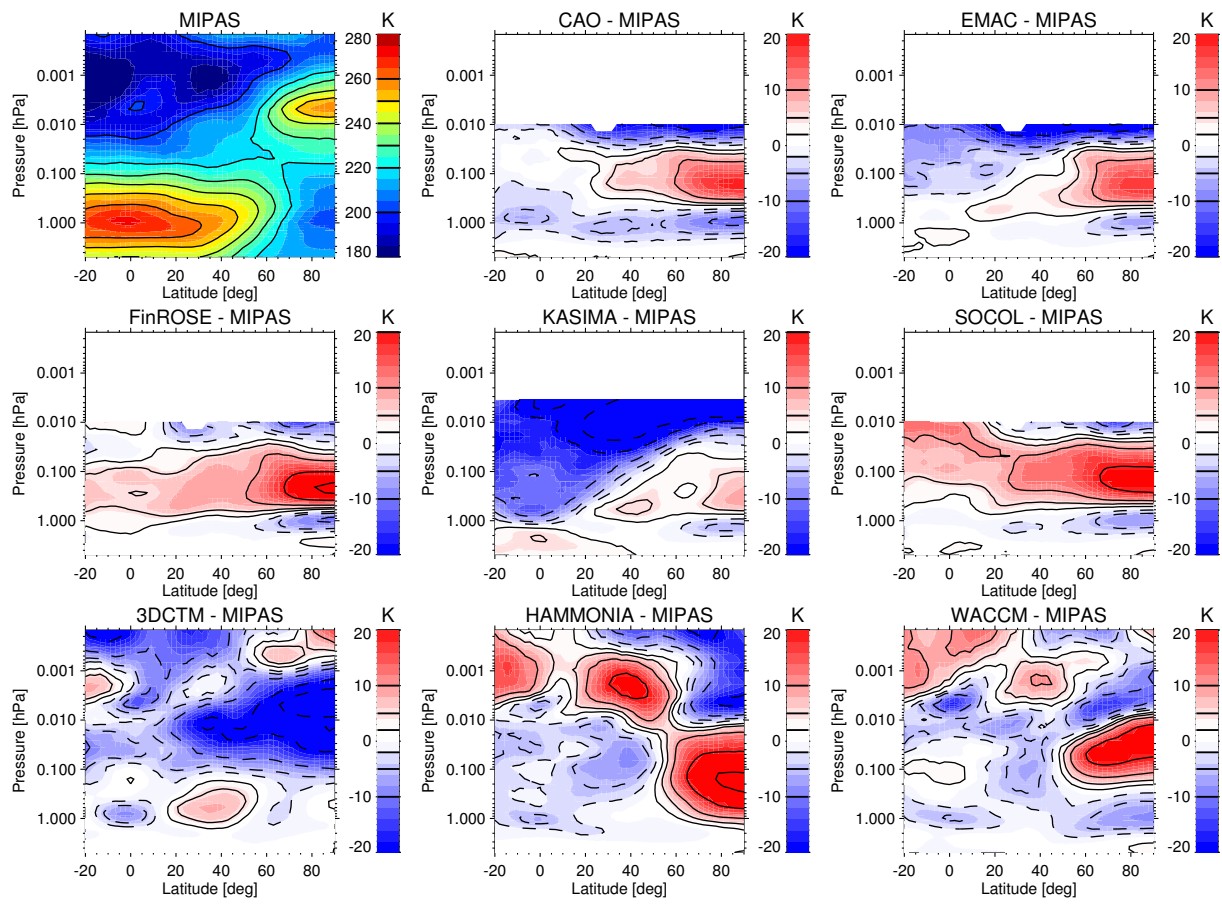

**Figure 20.** MIPAS-UA and modelled NH zonal mean temperature distribution on 15 February 2009.

important question which needs to be addressed in upcoming studies is the causes of the cold bias in the employed reanalysis datasets that have been found here.

The encountered spread of the magnitude of the ES-related $NO_x$ tongue below 0.1 hPa in the medium-top models, despite the prescription of a common odd nitrogen upper boundary above, deserves some further discussion. The consistency of simulated
5  temperature evolutions indicates that vertical transport is represented in these models in a similar way. It is therefore unlikely that differences in the descent velocities are the main cause for the spread. Differences in meridional transport and mixing above the vortex edge and subsequent enhanced photochemical loss could also contribute to the differences but would not explain overestimation. A most plausible explanation is the detailed treatment of the upper boundary condition. Prescribing at an altitude with too fast vertical transport, as indicated here at 0.2 hPa, will unavoidably cause a too strong flux of $NO_x$ into
10  the domain below. Therefore, models that use a UBC definition extending to lower pressure levels likely overestimate the $NO_x$ flux. This is, for example, the case of EMAC, which prescribes $NO_x$ in the entire vertical domain above 0.1 hPa: the peak




magnitude of the tongue is, as expected, close to the observations in the UBC domain. However, it becomes increasingly larger than the observed magnitude during the descent down to 0.7 hPa, where it is overestimated by a factor of 3. This highlights the importance of a realistic dynamical representation in the UBC domain in models prescribing $NO_x$ concentrations.

# 8   Conclusions

We have presented the results of the HEPPA-II intercomparison project, conducted in the framework of SPARC/WCRP's SOLARIS-HEPPA activity, which aims at evaluating the simulations of the NH polar winter 2008/2009 from eight atmospheric models by comparison with observations of temperature and concentrations of $NO_x$ and CO from seven satellite instruments. The large number of participating models allowed for a comprehensive assessment of the ability of state-of-the-art chemistry climate models to reproduce the observed EPP indirect effect in a dynamically perturbed NH winter under conditions of very low geomagnetic activity. The use of multi-instrument data for model evaluation not only allowed for the assessment of the significance of identified model biases, but also to estimate the uncertainty range of our current knowledge on tracer and temperature distributions in Arctic winters. It has been shown that the appropriate consideration of the instrument-specific sampling patterns is key to a meaningful multi-instrument analysis, particularly during perturbed dynamical conditions. The high degree of consistency between the comparisons of the models to individual observations has proven the reliability of the currently available satellite record during polar winter conditions.

Most models provide a good representation of the mesospheric tracer descent in general, and the EPP indirect effect in particular, during the unperturbed (pre-SSW) period of the NH winter 2008/2009. Observed $NO_x$ descent into the lower mesosphere and stratosphere is generally reproduced within 20%. Larger discrepancies of a few model simulations, resulting in overestimated $NO_x$ enhancements, could be traced back either to an unrealistic representation of the polar winter dynamics or to an inadequate prescription of the $NO_x$ partitioning at the uppermost model layer leading to boundary artefacts.

In March–April, after the ES event, however, modelled mesospheric and stratospheric $NO_x$ distributions deviate significantly from the observations. The too fast and early downward propagation of the $NO_x$ tongue, encountered in most simulations, coincides with a warm bias in the lower mesosphere (0.2–0.05 hPa) being likely caused by an overestimation of descent velocities. On the other hand, upper mesospheric temperatures at 0.05–0.001 hPa are in general underestimated by the high-top models after the onset of the ES event, being indicative of a too slow descent and hence too small $NO_x$ fluxes. As a consequence, the magnitude of the simulated $NO_x$ tongue is generally underestimated by these models. Descending $NO_x$ amounts simulated by the medium-top models with prescribed $NO_x$ are on average closer to the observations but show a large spread of up to several hundred percent. This is primarily attributed to the different vertical model regimes where the $NO_x$ upper bounder condition is applied.

In general, the intercomparison demonstrates the ability of state-of-the-art atmospheric models to reproduce the observed EPP indirect effect in dynamically and geomagnetically quiescent early NH winter conditions as present in November 2008 – January 2009. It should be noted, however, that the extrapolation of this result to high geomagnetic activity conditions should be done with caution since mid-energy electron impact in the mesosphere, which was of minor importance during this particular



winter, could lead to additional complications. Further, to obtain good agreement between simulated and observed mesospheric tracer descent it is ncessary to constrain stratospheric dynamics in the models by (re-)analysed meteorology.

The encountered differences between observed and simulated $NO_x$, CO, and temperature distributions during the perturbed phase of the 2009 NH winter (i.e., February – April), however, emphasize the need for model improvements in the dynamical representation of ES events in order to allow for a better description of the EPP indirect effect under these particular conditions. Our results reinforce the findings from previous studies that the adequate parametrisation of unresolved GWD, particularly of its non-orographic component, is crucial for achieving such improvements.

Many of the model-specific issues identified in the course of this project are currently being solved (e.g., Meraner et al., 2016). Lessons learned are hoped to be also of use for future model developments, particularly with respect to the consideration of EPP effects in upcoming coordinated model intercomparison projects. On the other hand, the encountered bias in the meteorological reanalysis data in the post-SSW upper stratosphere and lower mesosphere potentially triggered the common tendency of the models to produce a too early descent in the lower mesosphere. These results imply the need to improve data assimilation systems for producing reanalysis data, especially with respect to the representation of the polar winter upper stratosphere and mesosphere. This is particularly important because the use of specified dynamics in atmospheric models is a necessary step to allow for meaningful comparisons to observations on seasonal and shorter time scales.

## 9   Data availability

All the model and observational data supporting the analysis and conclusions have been archived and are available upon request from the corresponding author.

*Acknowledgements.* This work has been conducted in the frame of the WCRP/SPARC SOLARIS-HEPPA activity. The IAA team was supported by the Spanish MCINN under grant ESP2014-54362-P and EC FEDER funds. The MPI-MET team was supported by the Max-Planck-Gesellschaft (MPG), and computational resources were made available by Deutsches Klimarechenzentrum (DKRZ) through support from Bundesministerium für Bildung und Forschung (BMBF). The FMI team was supported by the Academy of Finland through the projects 276926 (SECTIC: Sun-Earth Connection Through Ion Chemistry), 258165, and 265005 (CLASP: Climate and Solar Particle Forcing). CAO team was supported by the Russian Science Foundation under grant 15-17-10024. SOCOL team was funded by Swiss National Science Foundation (SNSF) grants 200021-149182 (SILA), 200020-163206 (SIMA) and CRSII2-147659 (FUPSOL-II). S. Bender, M. Sinnhuber and H. Nieder (all KIT) gratefully acknowledge funding by the Helmholtz Association of German Research Centres (HGF), grant VH-NG-624. NCAR is sponsored by the National Science Foundation (NSF). Computing resources for WACCM simulations were provided by the Climate Simulation Laboratory at NCAR's Computational and Information Systems Laboratory, sponsored by the NSF and other agencies. The Atmospheric Chemistry Experiment (ACE), also known as SciSat, is a Canadian-led mission mainly supported by the Canadian Space Agency. Odin is a Swedish-led satellite project funded jointly by Sweden (SNSB), Canada (CSA), Finland (TEKES) and France (CNES), and is part of European Space Agency's (ESA) third party mission program.





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
