# Peer review of "HEPPA-II model-measurement intercomparison project: EPP indirect effects during the dynamically perturbed NH winter 2008–2009"

_Atmospheric Chemistry and Physics, 2016_

## Referee Comment (RC1) · Anonymous Referee #1 · 27 Dec 2016

This paper is an important and substantive contribution that, subject to my comments below, certainly merits publication in ACP. Its conclusions are credible and will be valuable for modelers. I do have lots of comments; their overall intent is to improve the paper and make it more useful to the community. A important editorial comment is that the paper is simply too long. There is a tradeoff that must be made between comprehensiveness and readability. These two parameters often anti-correlate. In this case, I believe the authors have leaned too far in the direction of comprehensiveness at the expense of readability. I also think there are some references that they should add-these are noted in the text below.

[Figure]

For example, Section 2 is too long by probably a factor of two. These datasets are mostly all mature and have been used for this problem in the past. For example, we do not need to be reminded that (line 31 on page 8) that SABER temperatures require non-LTE calculations in the retrievals or the details of when SABER is yawed north or that SABER's duty cycle is still nearly 100% (line 24-25). Rather, there is a rich literature of observational studies of these elevated stratopause events that could and should easily substitute for much of the text in Section 2. For example, although the paragraph on MLS is not too long, I am quite surprised that none of Gloria Manney's work was cited- for instance, her 2009 ACP paper which used the MLS data for the 2006 event much in the same fashion as presently done. Similarly, is there any usage here of ACE-FTS data that differs significantly from Randall et al., GRL, 2009 (also missing from the reference list)? And given that the Funke et al., 2014 JGR papers are cited, the usage of the MIPAS data should just cite those works; again, it is not necessary to tell us how non-LTE vibrational distributions were modeled (line 25, page 7). Finally, there must be dozens of papers which discuss SABER temperatures; two that might be useful to cite here are Siskind et al., GRL, 2007 for the 2006 event and Yamashita et al, JGR, 2013 (see their Figure 7 which shows 4 winters compared with MERRA). A similar comment concerns the models, although the situation is not as bad and Table 1 is useful. Nonetheless,, given the recent detailed discussions of HAMMONIA by Meraner et al and the discussions of WACCM by Randall et al. (2015), they should probably cut back their descriptions. For example, the citation to Meraner et al is sufficient to discuss HAMMONIA's gravity waves; all the information presented here on the source spectrum etc is superfluous and detracts from readability.

Other comments

1. I simply do not understand Figure 1. I don't see any symbol in Figure 1 which says "UBC". So how can the figure be showing it when none of the symbols do? What do the arrows show? I don't understand what the deviations are. Is this variance, standard deviation? What is the mathematic expression they are using? They should have two

panels- one for absolute values, and one for whatever these deviations are. Then, the text needs to discuss this carefully, not simply with some parenthetical reference.

2. Figure 3 should be deleted. It adds no new information that is not already clearly shown in Figure 2. I realize that the intent is to illustrate something about timing after the SSW- all I see is a jumble of points. Certainly the spread increases after Feb 1, but I cannot discern anything else.

3. One misunderstanding that I think I have concerns when exactly during the season does EPP-IE couple most strongly with the stratosphere? From reading Funke et al's two 2014 papers, it appears that significant NOx flux can penetrate into the stratosphere early in the winter, for example, November or December. Indeed in Funke et al., 2014, figure 10, one can see a tongue of EPP-NOy down dipping down to below 40 km on January 1st, much lower in altitude than the descending tongue in the post-SSW period. But in the present paper (page 30), it states that descending NOx can only be distinguished down to 0.3-0.5 hPa. This seems inconsistent and I think bears some explanation. Furthermore, if the pre-SSW NOx is more important for its contribution to the stratospheric NOx budget, then isn't the implication of the 2014 papers that the present focus on the post SSW descent is misplaced and of less relevance?

4. Section 7.1: It would be useful to discuss and justify the selection of CO as a tracer more. While I realize this is popular because of the low stratospheric values (page 27, line 5), I think it should also be stated that CH4 might be easier to simulate since it would not require the details of CO2 dissociation or reaction with OH to be handled so carefully. Indeed the present author has used CH4 (cf. his 2014 paper) as did Siskind in 2015 and Randall has used this for SH studies (her 2007 paper). Furthermore, with the selection of both CO and NOx we have two tracers that are being transported downgradient. Thus how can we know whether the transport we see is advective or diffusive? Is diffusion important in any of the simulations or in any of the model-model differences?

5. There is some discussion of the upper boundary that is used for the medium-top models. But the 2016 Funke paper states that one of its main objectives was to construct such a boundary. How do the values adopted here compare with what is presented in that paper?

6. Page 39: lines 12-13. I do not see where the consideration of the sampling patterns has been so important. Figure 19 shows that the temperatures are pretty similar. The only way this sentence can be justified if there were a figure which show a case where the sampling patter was not considered vs. a case where it was. I don't think they've done this. It would not detract from any of their conclusions if this sentence were deleted.

Editorial comments

1. Line 18, on page 7 seems strange. "MIPAS passes the equator in a southerly direction at 10:00 AM. . . . . . .observing the atmosphere day and night". Presumably the night time data are acquired when MIPAS passes the equator in a northerly direction? This is all phrased more tersely and more clearly in their 2014 JGR paper.

2. Page 25, line 1. The proper reference should be Siskind et al., JGR 2010 (not GRL, 2015), which discussed non-orographic drag in great depth. Likewise, consideration should be given to citing Chandran et al., GRL, 2011 who make this point as well.

3. Page 36, line 10, more grammar: Encountered is a verb and not an adjective and thus does not appear before the noun. It should read: "cold bias encountered at 1 hPa". Likewise page 38, line 3 "spread of the . . . . . . . encountered below 0.1 hPa". And again on page 40, line 3: ". . . encountered during the perturbed. . ." And finally on line 11, page 40.

---

## Referee Comment (RC2) · Anonymous Referee #2 · 23 Jan 2017

Overview:

The paper presents a comprehensive model intercomparison for perturbed and quiescent transport conditions in the mesosphere to evaluate the ability of models to capture EPP induced NOx transport and the EPP indirect effect. Models with high lids able to capture thermosphere NOx formation and lower lid models were included in this study. EPP and MLT NOx were introduced into models with lids lower than the mesopause via a boundary condition derived from MIPAS. The purpose of this study is to evaluate model transport characteristics that are relevant for EPP indirect effects on the

atmosphere. The 2008/2009 winter was chosen as the study period due in part to the availablity of an extensive set of satellite observations. This winter also had a major SSW in January which was associated with the descent of a substantial amount of NOx in spite of relatively quiet geomagnetic conditions. This is the only study to focus on EPP and NOx transport effects. Other model intercomparison projects have focused on dynamical characteristics during perturbed winters (e.g. Pedatella et al., 2014).

Aside from EMAC and FinROSE models were nudged with assimilated data (MERRA, ERA-Interim, ECMWF operational) below 1 hPa in order to approximate circulation during the observation period. This approach does not replace nudging by assimilated data in the mesosphere and is dictated by the lack of such data. It is hoped that having the model levels below the stratopause be in a circulation regime following observations imposes a dynamical boundary condition that propagates into the mesosphere and dominates the circulation response there as well. Although a non-local constraint on the dynamics will be imposed this way (e.g. via Rossby wave propagation), it will be partial since there are multiple solutions to a given set of boundary conditions. This "degeneracy" is apparent from non-orographic wave drag scheme tuning. It is possible to get reasonable temperatures and winds while not capturing tracer transport well.

Models are found to capture EPP NOx transport in the mesosphere during quiet periods reasonably well. However, the transport pattern in the wake of the major SSW in January 2009 is not captured well by models. Descent in models is too rapid and is associated with a warm bias in the lower mesosphere. At the same time high lid models underestimate downward transport in the upper mesosphere. Models with lids in the mesosphere that require an EPP NOx boundary condition show a large spread indicating substantial differences in vertical structure and transport in the mesosphere.

Aside from model weaknesses such as the tuning of the non-orgraphic gravity wave drag schemes, the assimilated wind and temperature data had inadequacies that likely contributed to the excessively early descent of NOx in the wake of the SSW (p 35–38).

[Figure]

Major Comments:

1) The authors raise the issue of assimilated data biases as a factor in the poor model performance at capturing transport in the wake of the major SSW in January. But another important issue is the use of nudging restricted below 1 hPa. The mesospheric circulation is not a linear problem and is not fully determined by the dynamical boundary condition at 1 hPa. In fact, multiple solutions are possible for the mesospheric state given the same set of stratopause conditions. In other words, it is not really possible to approximate nudging of the mesospheric circulation by restricting the circulation below 1 hPa. In the case of a CTM like FinROSE, the circulation needs to be specified in the mesosphere and the quality of the ECMWF data used is not clear and likely to be deficient (for example, gravity waves become increasingly important with altitude but are treated as noise in standard data assimilation systems.)

I would think that nudging restricted to below 1 hPa is a bigger problem than the biases in the assimilated data in the stratopause region. In particular, the non-orographic gravity wave drag schemes are essentially running in free mode in the mesosphere and impose biases since there is no local nudging to offset them. Non-orgraphic gravity wave drag cannot be constrained to the observed regime by imposing the right winds in the stratosphere (e.g., Scinocca and McLandress, 2005). Various built-in assumptions such as a constant source spectrum will still impact the response.

The basically free mode simulation of the mesosphere also affects the radiative transfer calculations. Radiatively active trace gas distributions are not likely to be ideally distributed to conform to observations. So the radiative damping impact on the evolution of the state in the wake of the major SSW will deviate from observations.

In addition the evolution of the highly nonlinear SSW in the mesosphere is not guaranteed to follow observations even if the wave amplitudes at the stratopause conform to observations. The progression of SSW events is rather complex (Matthewman et al., 2009) and the mesospheric component of a major SSW cannot be trivially constrained

by the stratospheric part. The SSW exhibits a high degree of top-down evolution with the mesosphere being an important layer of the atmosphere for this phenomenon.

It appears that relatively good performace of EMAC (Fig. 13, 14, 15, 16) and its application of assimilated data up to 0.2 hPa may not be a coincidence since every scrap of nudging in the mesosphere counts. FinROSE appears to do a better job as well (Fig., 13, 14) but peforms worse than EMAC apparently due to issues with the vertical upwelling. CTMs use geostrophic balance to infer the upwelling from the temperature and horizontal winds, but this fails to capture the ageostrophic part which is not negligible in the mesosphere due to the amplification of gravity waves and general deviation from geostrophic balance. In addition, several models linearly ramp nudging above 40 km so that they are not imposing as much of a constraint on the upper stratosphere and lower mesosphere dynamical state as models that fully nudge up to 1 hPa.

The authors mention the study of Siskind et al. (2015) which highlighted the need for nudging to much higher altitudes (92 km) to improve simulations of NO descent compared to observations. But there is no focused discussion of the limitations of the nudging approach adopted by most models in this intercomparison. Such a discussion has to be included either in the introduction or in the conclusions sections to put the results from these models in a better context.

References:

N. J. Matthewman, J.G. Esler, A.J. Charlton-Perez and L.M. Polvani: A new look at stratospheric sudden warmings. Part III. Polar vortex evolution and vertical structure, J. Climate, 22, 1566-1585 (2009).

J. Scinocca, and C. McLandress: The GCM Response to Current Parameterizations of Nonorographic Gravity Wave Drag, J. Atmos. Sci., 62(7), 2394-2412, 10.1175/JAS3483.1, 2005.

---

## Author Comment (AC1) · 7 Feb 2017

**Responses to the comments of Referee #1**

*We thank Referee #1 for the thoughtful comments and suggestions, which certainly helped to improve the clarity of the manuscript. Please find below our detailed point-by-point reply to the comments, which we hope have addressed all satisfactorily.*

This paper is an important and substantive contribution that, subject to my comments below, certainly merits publication in ACP. Its conclusions are credible and will be valuable for modelers.

*Thank you very much!*

I do have lots of comments; their overall intent is to improve the paper and make it more useful to the community. An important editorial comment is that the paper is simply too long. There is a tradeoff that must be made between comprehensiveness and readability. These two parameters often anti-correlate. In this case, I believe the authors have leaned too far in the direction of comprehensiveness at the expense of readability. I also think there are some references that they should add- these are noted in the text below.

*Reply: See detailed responses below.*

For example, Section 2 is too long by probably a factor of two. These datasets are mostly all mature and have been used for this problem in the past. For example, we do not need to be reminded that (line 31 on page 8) that SABER temperatures require non-LTE calculations in the retrievals or the details of when SABER is yawed north or that SABER's duty cycle is still nearly 100% (line 24-25). Rather, there is a rich literature of observational studies of these elevated stratopause events that could and should easily substitute for much of the text in Section 2. For example, although the paragraph on MLS is not too long, I am quite surprised that none of Gloria Manney's work was cited- for instance, her 2009 ACP paper which used the MLS data for the 2006 event much in the same fashion as presently done. Similarly, is there any usage here of ACE-FTS data that differs significantly from Randall et al., GRL, 2009 (also missing from the reference list)? And given that the Funke et al., 2014 JGR papers are cited, the usage of the MIPAS data should just cite those works; again, it is not necessary to tell us how non-LTE vibrational distributions were modeled (line 25, page 7). Finally, there must be dozens of papers which discuss SABER temperatures; two that might be useful to cite here are Siskind et al., GRL, 2007 for the 2006 event and Yamashita et al, JGR, 2013 (see their Figure 7 which shows 4 winters compared with MERRA).

*Reply: We agree that Section 2 is rather long and can be shortened. In the revised version we will remove those details on the instruments and retrievals that are available elsewhere. However, we think that some details are relevant for this paper and should be maintained even if provided in previous work. This is the case for the information on sampling patterns and data gaps during the period of interest, as well as on accuracies and known biases.*

*We will further add the mentioned references for previous observational studies*

*about elevated stratopause events, based on the same instruments, in order to better put our work into the context of existing work. Note, however, that these studies employed different data versions compared to those used in the present work.*

A similar comment concerns the models, although the situation is not as bad and Table 1 is useful. Nonetheless,, given the recent detailed discussions of HAMMONIA by Meraner et al and the discussions of WACCM by Randall et al. (2015), they should probably cut back their descriptions. For example, the citation to Meraner et al is sufficient to discuss HAMMONIA's gravity waves; all the information presented here on the source spectrum etc is superfluous and detracts from readability.

*Reply: The model descriptions will be shortened (in particular, the information already listed in Table 1 will not be repeated again in the text). However, we think that detailed information on the non-orographic GW parameterizations should be maintained, as it is relevant for the paper and it helps to understand the individual model results.*

Other comments

1. I simply do not understand Figure 1. I don't see any symbol in Figure 1 which says "UBC". So how can the figure be showing it when none of the symbols do? What do the arrows show? I don't understand what the deviations are. Is this variance, standard deviation? What is the mathematic expression they are using? They should have two panels- one for absolute values, and one for whatever these deviations are. Then, the text needs to discuss this carefully, not simply with some parenthetical reference.

*Reply: We apologise that Figure 1 of the discussion paper is difficult to understand. The plot shows daily averaged NOx mixing ratios from satellite observations and those of the upper boundary condition sampled at the respective observations' time and location. We have improved both figure and caption for the revised version to make this clearer (see new figure below).*

[Figure]

**Figure 1.** Upper panel: Daily averaged $NO_x$ mixing ratios from satellite observations (open squares) at 0.022 hPa within 60–90°N (black=MIPAS-NOM, blue = MIPAS-UA, red= SMR/Odin, green = ACE-FTS) and those of the upper boundary condition (filled diamonds) sampled at the respective observations' time and location. Lower panel: average latitude of observations. All averages are area-weighted.

2. Figure 3 should be deleted. It adds no new information that is not already clearly shown in Figure 2. I realize that the intent is to illustrate something about timing after the SSW- all I see is a jumble of points. Certainly the spread increases after Feb 1, but I cannot discern anything else.

*Reply: Figure 3 will be deleted in the revised version.*

3. One misunderstanding that I think I have concerns when exactly during the season does EPP-IE couple most strongly with the stratosphere? From reading Funke et al's two 2014 papers, it appears that significant NOx flux can penetrate into the stratosphere early in the winter, for example, November or December. Indeed in Funke et al., 2014, figure 10, one can see a tongue of EPP-NOy down dipping down to below 40 km on January 1st, much lower in altitude than the descending tongue in the post-SSW period. But in the present paper (page 30), it states that descending NOx can only be distinguished down to 0.3-0.5 hPa. This seems inconsistent and I think bears some explanation.

*Reply: The Funke et al. (2014) paper discusses EPP-NOy (the contribution of EPP-generated NOy to total NOy), while in the present study we analyse NOx. Since NOx is converted to other NOy species (HNO3) below approximately 45 km, the NOy descent below this altitude cannot be traced by NOx. Further, dilution of descending NOx with the stratospheric background masks the descent below 45 km in the present analysis, while this is not the case when looking at EPP-NOy. This will be mentioned in the revised version.*

Furthermore, if the pre-SSW NOx is more important for its contribution to the stratospheric NOx budget, then isn't the implication of the 2014 papers that the present focus on the post SSW descent is misplaced and of less relevance?

*Reply: The higher relevance of the pre-SSW NOx descent for the stratospheric NOy budget is particularly the reason why we included it in our analysis at a similar level of detail as the post-SSW descent, though larger deviations between models and observations in the latter case made it necessary to extend the discussion of post-SSW descent.*

4. Section 7.1: It would be useful to discuss and justify the selection of CO as a tracer more. While I realize this is popular because of the low stratospheric values (page 27, line 5), I think it should also be stated that CH4 might be easier to simulate since it would not require the details of CO2 dissociation or reaction with OH to be handled so carefully. Indeed the present author has used CH4 (cf. his 2014 paper) as did Siskind in 2015 and Randall has used this for SH studies (her 2007 paper). Furthermore, with the selection of both CO and NOx we have two tracers that are being transported downgradient. Thus how can we know whether the transport we see is advective or diffusive? Is diffusion important in any of the simulations or in any of the model-model differences?

*Reply: Indeed we have analysed CH4 in a similar manner as CO (see Figure below), however, we decided not to include these results in the paper for the following*

*reasons:*

- *Modelled CH4 distributions deviate larger from each other and from the observations than CO does, likely because of differences in the simulated chemical losses, as well as in the transport by the Brewer Dobson circulation. Despite of this, the evolution of CH4 during the ES event behaves qualitatively similar (though with opposite vertical gradient) as CO such that not much new is learned.*

- *CH4 concentrations drop rapidly towards higher altitudes such that observations are getting typically below the noise level above 0.1 hPa. The useful vertical range is thus much reduced compared to CO.*

- *The use of a second tracer that is being transported downgradient (in addition to NOx) is intentional. Since CO has no EPP source, it allows us to assess whether model biases are caused by deficiencies in the transport scheme or in the NOx production scheme (or UBC implementation, in the case of medium-top models).*

*Regarding the question whether diffusive or advective transport dominates, we would expect that in the case of a significant diffusive contribution the CO peak would be shifted slightly downwards while the CH4 minimum would be shifted upwards. A qualitative comparison of the figure below and Figure 10 of the manuscript suggests that this is not the case, neither in the observations nor in the models.*

[Figure]

5. There is some discussion of the upper boundary that is used for the medium-top models. But the 2016 Funke paper states that one of its main objectives was to construct such a boundary. How do the values adopted here compare with what is presented in that paper?

*Reply: The UBC employed in the present study is based on the same MIPAS observations as used in Funke et al. (2016) for the construction of their semi-empirical model. Their Figure 8 already compares the semi-empirical UBC with the observed NOy (used here as UBC). The purpose of the semi-empirical model is to provide a NOy UBC for long-term climate simulations or for shorter simulations of periods not covered by MIPAS observations. Since this is not the case in the present study, we decided not to use the semi-empirical model but to rely on the observed NOx.*

6. Page 39: lines 12-13. I do not see where the consideration of the sampling patterns has been so important. Figure 19 shows that the temperatures are pretty similar. The only way this sentence can be justified if there were a figure which show a case where the sampling pattern was not considered vs. a case where it was. I don't think they've done this. It would not detract from any of their conclusions if this sentence were deleted.

*Reply: We do not fully agree, particularly with respect to the sampling impact on the NOx comparisons. In our opinion, there are significant differences, especially for the ACE sampling. For instance, we show the same model NOx sampled at different locations and times in Figure 2. Here, the ES-related tongue as observed by ACE has apparently a different timing compared to the other instruments. This is caused by the seasonal propagation of latitudes sounded by ACE (see also Fig.1, lower panel) and needs to be considered when comparing to other observations or models.*

Editorial comments

1. Line 18, on page 7 seems strange. "MIPAS passes the equator in a southerly direction at 10:00 AM. . .. . . .observing the atmosphere day and night". Presumably the nighttime data are acquired when MIPAS passes the equator in a northerly direction? This is all phrased more tersely and more clearly in their 2014 JGR paper.

*Reply: Following the major first comment above, this sentence will be removed.*

2. Page 25, line 1. The proper reference should be Siskind et al., JGR 2010 (not GRL, 2015), which discussed non-orographic drag in great depth. Likewise, consideration should be given to citing Chandran et al., GRL, 2011 who make this point as well.

*Reply: Both references will be cited adequately.*

3. Page 36, line 10, more grammar: Encountered is a verb and not an adjective and thus does not appear before the noun. It should read: "cold bias encountered at 1 hPa". Likewise page 38, line 3 "spread of the . . .. . . .. encountered below 0.1 hPa". And again on page 40, line 3: ". . . encountered during the perturbed. . ." And finally on line 11, page 40.

*Reply: Thank you very much for the grammar's corrections. These sentences will be corrected.*

---

## Author Comment (AC2) · 7 Feb 2017

**Responses to the comments of Referee #2**

*We thank Referee #2 for the thoughtful comments and suggestions, which certainly helped to improve the clarity of the manuscript. Please find below our detailed point-by-point reply to the comments, which we hope have addressed all satisfactorily.*

1) The authors raise the issue of assimilated data biases as a factor in the poor model performance at capturing transport in the wake of the major SSW in January. But another important issue is the use of nudging restricted below 1 hPa. The mesospheric circulation is not a linear problem and is not fully determined by the dynamical boundary condition at 1 hPa. In fact, multiple solutions are possible for the mesospheric state given the same set of stratopause conditions. In other words, it is not really possible to approximate nudging of the mesospheric circulation by restricting the circulation below 1 hPa. In the case of a CTM like FinROSE, the circulation needs to be specified in the mesosphere and the quality of the ECMWF data used is not clear and likely to be deficient (for example, gravity waves become increasingly important with altitude but are treated as noise in standard data assimilation systems.)

*Reply: We agree that restricting the nudging of the models to below 1 hPa leaves the mesospheric circulation largely unconstrained, thus making it more difficult to compare to observations. However, the major aim of the paper is to evaluate the capacity of climate models to simulate a realistic polar winter NOx descent. This could hardly be assessed if the mesospheric circulation was entirely constrained by nudging. The restriction of specified dynamics to the stratosphere is a compromise that is hoped to provide a realistic evolution of mesospheric meteorology by upward control, while still allowing for the assessment of self-generated tracer descent in the models.*

I would think that nudging restricted to below 1 hPa is a bigger problem than the biases in the assimilated data in the stratopause region. In particular, the non-orographic gravity wave drag schemes are essentially running in free mode in the mesosphere and impose biases since there is no local nudging to offset them. Non-orographic gravity wave drag cannot be constrained to the observed regime by imposing the right winds in the stratosphere (e.g., Scinocca and McLandress, 2005). Various built-in assumptions such as a constant source spectrum will still impact the response.

*Reply: Yes, we agree. But, as mentioned by the reviewer, the GWD schemes are running basically in a free-running mode and therefore one would expect a more random-like spread of model results. However, our results suggest a systematic model bias with respect to the temperature distribution and the timing of NOx descent in the lower mesosphere. How can such a systematic behaviour be explained by the lack of nudging above 1 hPa?*

The basically free mode simulation of the mesosphere also affects the radiative transfer calculations. Radiatively active trace gas distributions are not likely to be

ideally distributed to conform to observations. So the radiative damping impact on the evolution of the state in the wake of the major SSW will deviate from observations.

*Reply: This is a good point! The potential impact of tracer distributions on the radiative cooling/heating will be mentioned in the revised version.*

In addition the evolution of the highly nonlinear SSW in the mesosphere is not guaranteed to follow observations even if the wave amplitudes at the stratopause conform to observations. The progression of SSW events is rather complex (Matthewman et al., 2009) and the mesospheric component of a major SSW cannot be trivially constrained by the stratospheric part. The SSW exhibits a high degree of top-down evolution with the mesosphere being an important layer of the atmosphere for this phenomenon.

*Reply: We agree that SSWs exhibit a high degree of top-down evolution. However, we'd like to mention in this context that it is presently not known if the mesosphere has any control on the stratosphere or, if the mesospheric SSW precursors are triggered from below.*

It appears that relatively good performance of EMAC (Fig. 13, 14, 15, 16) and its application of assimilated data up to 0.2 hPa may not be a coincidence since every scrap of nudging in the mesosphere counts.

*Reply: EMAC applies the NOx UBC down to 0.1 hPa. This represents a trivial reason for a better performance with respect to the representation of NOx in the region above this level. Note that this model significantly overestimates the descending NOx amounts in the post-SSW phase below 0.2 hPa (see Figure 17) despite the application of assimilated data in this region.*

FinROSE appears to do a better job as well (Fig., 13, 14) but performs worse than EMAC apparently due to issues with the vertical upwelling.

*Reply: We'd like to point out that the model performance with respect to NOx descent in the pre-SSW phase (Figures 13 and 14) is generally good (except for those models with identified issues regarding the UBC implementation or missing unresolved GWD).*

CTMs use geostrophic balance to infer the upwelling from the temperature and horizontal winds, but this fails to capture the ageostrophic part which is not negligible in the mesosphere due to the amplification of gravity waves and general deviation from geostrophic balance.

*Reply: We agree with this view.*

In addition, several models linearly ramp nudging above 40 km so that they are not imposing as much of a constraint on the upper stratosphere and lower mesosphere dynamical state as models that fully nudge up to 1 hPa.

*Reply: Also agreed. We will mention this explicitly in the revised version.*

The authors mention the study of Siskind et al. (2015) which highlighted the need for nudging to much higher altitudes (92 km) to improve simulations of NO descent compared to observations. But there is no focused discussion of the limitations of the nudging approach adopted by most models in this intercomparison. Such a discussion has to be included either in the introduction or in the conclusions sections to put the results from these models in a better context.

*Reply: Such a discussion will be added.*

References:

N. J. Matthewman, J.G. Esler, A.J. Charlton-Perez and L.M. Polvani: A new look at stratospheric sudden warmings. Part III. Polar vortex evolution and vertical structure, J. Climate, 22, 1566-1585 (2009).

J. Scinocca, and C. McLandress: The GCM Response to Current Parameterizations of Nonorographic Gravity Wave Drag, J. Atmos. Sci., 62(7), 2394-2412, 10.1175/JAS3483.1, 2005.